Manuscript prepared for Atmos. Chem. Phys.
with version 2014/09/16 7.15 Copernicus papers of the LaTeX class copernicus.cls.
Date: 8 December 2018

# Representing sub-grid scale variations in nitrogen deposition associated with land use in a global Earth System Model: implications for present and future nitrogen deposition fluxes over North America

Fabien Paulot[1,2], Sergey Malyshev[1], Tran Nguyen[3], John D. Crounse[4],
Elena Shevliakova[1], and Larry W. Horowitz[1]

[1]Geophysical Fluid Dynamics Laboratory, National Oceanic and Atmospheric Administration,
Princeton, New Jersey, USA
[2]Program in Atmospheric and Oceanic Sciences, Princeton University, New Jersey, USA
[3]Department of Environmental Toxicology, UC Davis, Davis, California, USA
[4]Division of Geological and Planetary Sciences, Caltech, Pasadena, California, USA

*Correspondence to:* Fabien.Paulot@noaa.gov

**Abstract.** Reactive nitrogen (N) emissions have increased over the last 150 years as a result of greater fossil fuel combustion and food production. The resulting increase in N deposition can alter the function of ecosystems, but characterizing its ecological impacts remains challenging, in part because of uncertainties in model-based estimates of N dry deposition. Here, we use the Geophysical

Fluid Dynamics Laboratory (GFDL) atmospheric chemistry-climate model (AM3) coupled with the GFDL land model (LM3) to estimate dry deposition velocities. We leverage the tiled structure of LM3 to represent the impact of physical, hydrological, and ecological heterogeneities on the surface removal of chemical tracers. We show that this framework can be used to estimate N deposition at more ecologically-relevant scales (e.g., natural vegetation, water bodies) than from the coarse-

resolution global model AM3. Focusing on North America, we show that the faster removal of N over forested ecosystems relative to cropland and pasture implies that coarse-resolution estimates of N deposition from global models systematically underestimate N deposition to natural vegetation by 10 to 30% in the Central and Eastern US. Neglecting the sub-grid scale heterogeneity of dry deposition velocities also results in an underestimate (overestimate) of the amount of reduced

(oxidized) nitrogen deposited to water bodies. Overall, changes in land cover associated with human activities are found to slow down the removal of N from the atmosphere, causing a reduction in the dry oxidized, dry reduced, and total (wet+dry) N deposition over the contiguous US of 8%, 26%, and 6%, respectively. We also find that the reduction in the overall rate of removal of N associated with land-use change tends to increase N deposition on the remaining natural vegetation

and facilitate N export to Canada. We show that sub-grid scale differences in the surface removal

of oxidized and reduced nitrogen imply that projected near-term (2010–2050) changes in oxidized (-47%) and reduced (+40%) US N emissions will cause opposite changes in N deposition to water bodies (increase) and natural vegetation (decrease) in the Eastern US, with potential implications for acidification and ecosystems.

## 1 Introduction

Fossil fuel combustion and food production release reactive nitrogen (N) to the atmosphere (Fowler et al., 2013). Once in the atmosphere, N can be transported over long distances before it is removed by dry and wet deposition, providing greater N inputs to otherwise pristine regions (e.g., national parks, boreal forests) (Paulot et al., 2014; Malm et al., 2016). Since N can be a limiting nutrient, the increase in N deposition may promote ecosystem productivity, (Townsend et al., 1996; Magnani et al., 2007; Pregitzer et al., 2008; Reay et al., 2008; Dezi et al., 2010; Wårlind et al., 2014; Devaraju et al., 2015) especially in boreal regions (Högberg, 2012; Gundale et al., 2014; Fleischer et al., 2015). Increasing N deposition can also cause adverse environmental impacts for terrestrial ecosystems including soil acidification, loss of biodiversity, and eutrophication (Stevens et al., 2004; Bobbink et al., 2010; Sutton et al., 2011; Pardo et al., 2011; Sheppard et al., 2011; Phoenix et al., 2012; Erisman et al., 2013; de Vries et al., 2015; Simkin et al., 2016). In the US, oxidized N deposition is projected to decrease as a result of effective controls on NO emissions, but deposition of reduced N ($NH_x \equiv NH_3 + NH_4^+$), primarily from agricultural emissions of $NH_3$, is projected to remain elevated or even increase (Dentener et al., 2006; Ellis et al., 2013; Paulot et al., 2013; Lamarque et al., 2013; Li et al., 2016). This raises concerns of irreversible damages to sensitive biomes (Pardo et al., 2011; Meunier et al., 2016; Grizzetti, 2011; Dise, 2011), such as high-elevation lakes (Wolfe et al., 2003; Baron et al., 2012; Lepori and Keck, 2012), and organisms (e.g., lichen (Johansson et al., 2012)).

Significant challenges remain in quantifying the long-term impacts of N deposition on ecosystems in a changing climate (Sutton et al., 2008; Wu and Driscoll, 2010; Phoenix et al., 2012; Högberg, 2012; de Vries et al., 2015; Storkey et al., 2015), including uncertainties in the speciation, magnitude and spatial distribution of the N deposition flux itself (Sutton et al., 2008; Ochoa-Hueso et al., 2011; Jickells et al., 2013; Fleischer et al., 2013). Many approaches have been developed to provide high-resolution, ecosystem-relevant estimates of both wet and dry N deposition, including statistical models (Singles et al., 1998; Dore et al., 2007; Weathers et al., 2006; Dore et al., 2012), high-resolution nested chemical transport model ($\simeq 4 \times 4\,km$ (Vieno et al., 2009; Simkin et al., 2016)), and hybrid approaches that combine high-resolution regional chemical transport models with observed N fluxes and atmospheric concentrations (e.g. using the Community Multiscale Air Quality Modeling System (Schwede and Lear, 2014; Bytnerowicz et al., 2015; Williams et al., 2017)). However, the elevated computational requirement associated with high-resolution atmospheric models make such approaches impractical for assessing the long-term impact of N deposition on ecosystems, its

sensitivity to climate change, and ultimately its coupling with the carbon cycle (Smith et al., 2014; Zaehle et al., 2010; Fleischer et al., 2013; Dirnböck et al., 2017; Fleischer et al., 2015). For such questions, estimates of N deposition are generally derived from global models with coarse resolution ($\simeq$ 100km, (Dentener et al., 2006; Lamarque et al., 2013)). This introduces a large uncertainty

(Hertel, 2011) in N deposition estimates especially for dry deposition, which can vary over short distances ($\sim$1 km) in response to changes in the physical, hydrological, and ecological state of the surface (Weathers et al., 2000; Hicks, 2006, 2008; De Schrijver et al., 2008; Ponette-González et al., 2010; Templer et al., 2014; Tulloss and Cadenasso, 2015).

    The goal of this study is to develop a framework to diagnose ecosystem-specific N dry deposi-

tion fluxes within a global chemistry climate model on decadal to centennial time scales. First we describe the coupling of the Geophysical Fluid Dynamics Laboratory (GFDL) land-model (LM3) to the GFDL atmospheric chemistry–climate model (AM3) to represent the impact of natural (e.g., vegetation type, soil and canopy wetness) and man-made (e.g., deforestation, cropping) heterogeneities on dry deposition. We then show that the tiled structure of LM3 can be leveraged to derive N depo-

sition on a more ecologically-relevant scale (e.g., deposition on water bodies or natural vegetation). Finally, we discuss how this framework can be used to better represent the impact of land-use change and future trends in N emissions on N deposition.

## 2   Methods

### 2.1   Model description

We use an updated version of the GFDL AM3 atmospheric chemistry–climate model (Donner et al., 2011; Naik et al., 2013; Paulot et al., 2016) to simulate atmospheric dynamics and chemistry. Except for the treatment of dry deposition, the model configuration is identical to the one recently described by Paulot et al. (2016) and Paulot et al. (2017), including updates to wet deposition and the chemistry of sulfate and nitrate. The horizontal resolution of the model is 200km with 48 vertical levels.

In AM3, the surface removal of chemical tracers is calculated using a prescribed monthly climatology of dry deposition velocities (Naik et al., 2013; Paulot et al., 2016). The lack of a dynamic representation of dry deposition reduces the ability of the model to capture the impact of past and future variability in environmental conditions (e.g., drought (Wu et al., 2016), climate change) and land-use change on atmospheric chemistry. We note that these limitations are not specific to AM3

but affect all chemical transport models that do not include a comprehensive land model (Ellis et al., 2013; Ran et al., 2017).

    Here, we describe the development of a new model, in which dry deposition of gaseous and aerosol species is calculated within the dynamic vegetation model LM3 (Shevliakova et al., 2009; Milly et al., 2014). The combined model will be referred to as AM3-LM3-DD hereafter.

LM3 is a comprehensive climate land model that includes detailed representations of vegetation dynamics and hydrology and is designed to be run over decadal to century time scales under both historical and future conditions. LM3 can be run both coupled with AM3 and in standalone mode with prescribed meteorological fields (Milly et al., 2014).

In LM3, the heterogeneity of the land surface and vegetation is represented using a sub-grid mo-
saic of tiles (Shevliakova et al., 2009; Malyshev et al., 2015) as illustrated in Fig. 1. Each tile has distinct energy and moisture balances for a vegetation–snow–soil column, biophysical properties, and exchanges of radiant and turbulent fluxes with the overlying atmosphere. LM3 predicts physical, biogeochemical, and ecological characteristics for each sub-grid land surface tile from the top of the vegetation canopy to the bottom of the soil column, including leaves and canopy temperature,
canopy-air specific humidity, stomatal conductance, snow cover and depth, runoff, vertical distribution of soil moisture, ice, and temperature. The land-use history is prescribed from the Hurtt et al. (2011) reconstruction for each grid cell in terms of annual transition rates among four distinct land-use types: undisturbed (hereafter referred to as natural), crops, pastures, and secondary vegetation. Secondary vegetation is defined in LM3 as the vegetation recovering after land-use and land-cover
changes and not currently managed. This includes all abandoned agricultural land as well as the land where wood was harvested at least once in prior years. The model keeps track of different recovery states by creating a secondary vegetation tile every time a disturbance occurs and simulating the subsequent vegetation regrowth in the tile. To avoid unrestricted growth of the number of tiles, the number of secondary vegetation tiles is limited to 10 per grid cell in the configuration of LM3 used
here. When more than 10 secondary vegetation titles exist in a grid cell, secondary vegetation tiles with similar properties are merged (Shevliakova et al., 2009), while preserving water, energy, and carbon balances. Land properties that affect the surface removal of chemical tracers, such as snow cover, canopy liquid water and snow mass, surface and canopy temperature, leaf area index (LAI), stomatal conductance, and vegetation height are all prognostic (Shevliakova et al., 2009). Vegetation
carbon is partitioned into five pools: leaves, fine roots, sapwood, heartwood, and labile storage. The model simulates changes in vegetation and soil carbon pools, as well as the carbon exchange among these pools and the atmosphere. The sizes of the pools are modified daily depending on the carbon uptake according to a set of allocation rules. Additionally, the model simulates changes in the vegetation carbon pools due to phenological processes, natural mortality, and fire. LAI is determined by
vegetation leaf biomass and specific leaf area, prescribed for each vegetation type. Each vegetated tile has a unique vegetation type (C3 grass, C4 grass, temperate deciduous, coniferous, or tropical vegetation), which is determined based on biogeographical rules that take into account environmental conditions as well as vegetation biomass in each tile (Shevliakova et al., 2009). The fraction of the canopy covered by liquid water ($f_l$) and snow ($f_s$) are estimated from the intercepted canopy

liquid water mass ($w_l$) and snow mass ($w_s$) following Bonan (1996):

$$f_i = \left(\frac{w_i}{W_{i,max}}\right)^{\frac{2}{3}} \qquad i \in \{l,s\} \tag{1}$$

where $W_{l,max} = 0.02\,kg\,m^{-2}$ and $W_{s,max} = 0.2\,kg\,m^{-2}$ are the maximum liquid water and snow holding capacities, respectively. If both snow and liquid water are present simultaneously, water and snow are assumed to be distributed independently of each other.

The representation of management practices is important in determining the impact of land-use change on dry deposition, as it affects the vegetation type, and the seasonality of the vegetation cover. In LM3, crop harvesting and pasture grazing are performed annually at the end of the calendar year (Malyshev et al., 2015). Previous work has shown that this treatment contributes to an underestimate of the impact of management on land cover (Malyshev et al., 2015). To address these biases, we make the following modifications. For pasture, we assume that 10% of leaf biomass removed daily by grazing, provided LAI exceeds 2 to avoid overgrazing. This higher grazing frequency and intensity are needed to avoid the excessive growth of vegetation biomass on pasture in the tropics and mid latitudes, a problem which was noted in previous versions of LM3 (Malyshev et al., 2015) leading to misclassification of pasture vegetation cover as forests (Malyshev et al., 2015). LM3 does not estimate the cropping schedule (e.g., Bondeau et al. (2007)), so we specify planting and harvesting dates from the global monthly irrigated and rainfed crop areas climatology (Portmann et al., 2010). The impact of management practices on the timing and magnitude of agricultural emissions (e.g., Paulot et al. (2014)) is not accounted for in AM3-LM3-DD.

The tiled structure of LM3 is especially useful to diagnose fluxes to areas, such as natural vegetation or water bodies, which are generally not well-represented by the average properties of the grid-box, in which they are located, because of their small geographical extent (Fig. S1).

The dry deposition velocity ($v_d(X)$) for species X is calculated independently for each tile following the widely used electrical circuit analogy (Fig. 1) (Hicks et al., 1987; Wesely, 1989; Zhang et al., 2001, 2003).

$$v_d(X) = \left[ R_a + \cfrac{1}{\cfrac{1}{R_{ac,g} + R_{b,g}(X) + R_{sf,g}(X)} + \cfrac{1}{R_{ac,v} + \cfrac{1}{[R_{b,s} + R_{sf,s}]^{-1} + \cfrac{1}{R_{b,v} + \left[R_{sf,v}^{-1} + (R_m + R_s)^{-1}\right]^{-1}}}}} \right]^{-1} \tag{2}$$

Briefly, the aerodynamic resistance ($R_a$) to the exchange of tracers between the canopy and the atmosphere is determined using Monin-Obukhov similarity theory. Within the canopy, the aerodynamic resistances to the ground ($R_{ac,g}$) and to the vegetation ($R_{ac,v}$) are independent of the chemical tracer and taken from Erisman (1994) and Choudhury and Monteith (1988), respectively.

$$R_{ac,g} = \frac{14(LAI + SAI)h}{u_\star} \tag{3}$$

$$R_{ac,v} = \frac{1}{(LAI + SAI) \cdot g_b} \text{ with } g_b = 0.01(1 - exp(-3/2)/3)\sqrt{V} \tag{4}$$

where $SAI$, $h$, V, and $u_\star$ are the stem area index (unitless), the height of the vegetation (in m), the normalized wind (m/s) at the top of the canopy, and the friction velocity (m/s), respectively. Note that
unlike Erisman (1994), we include SAI in the calculation of $R_{ac,g}$, which tends to reduce deposition to the ground in winter.

We focus next on the representation of the dry deposition of gases, which is much faster than that of fine particles (Zhang et al., 2002).

Following Jensen and Hummelshøj (1995) and Jensen and Hummelshøj (1997), the canopy lami-
nar resistance ($R_{b,v}$) is defined as:

$$R_{b,v}(X) = \frac{1}{D_X} \left( \frac{u_\star}{\nu} LAI \right)^{-2/3} (100lw)^{1/3} \tag{5}$$

where $lw$ the characteristic obstacle length of the canopy (in m, Table S1), $\nu$ the kinematic viscosity, and $D_X$ the diffusivity of species $X$. Following Hicks et al. (1987), the stem laminar resistance is:

$$R_{b,s}(X) = \frac{2}{\kappa u_\star} \left( \frac{Sc(X)}{Pr} \right)^{2/3} \tag{6}$$

where $Pr$ is the Prandtl number, $Sc(X)$ is the Schmidt number, the ratio of the kinematic to the mass diffusivity ($Sc \propto D_X{}^{-1}$), $\kappa$ the von Karman constant ($\kappa = 0.4$). Similarly, the ground surface laminar resistance is:

$$R_{b,g}(X) = \frac{2}{\kappa u_{g\star}} \left( \frac{Sc(X)}{Pr} \right)^{2/3} \tag{7}$$

where $u_{g\star}$ is the friction velocity near the ground (Loubet et al., 2006).
The mesophyll resistance is expressed following Wesely (1989):

$$R_m(X) = \left( 10^5/3000 \cdot \alpha(X) + 100 \cdot \beta(X) \right)^{-1} \tag{8}$$

The stomatal resistance ($R_s(X)$) is calculated as

$$R_s(X) = \sqrt{\frac{M(X)}{M(\text{H}_2\text{O})}} R_s(\text{H}_2\text{O}) \tag{9}$$

where $M(X)$ is the molecular weight of species $X$ and $R_s(\text{H}_2\text{O})$ is the stomatal resistance for water
vapor, calculated according to the Leuning model (Leuning, 1995; Milly et al., 2014). This model

accounts for the impact of water stress and $CO_2$ concentration, which have been shown to modulate the response of surface ozone to drought (Huang et al., 2016) and $CO_2$ increase (Sanderson et al., 2007). Cuticle ($v$), stem ($s$), and ground ($g$) resistances for species $X$ are parameterized based on $SO_2$ and $O_3$:

$$R_{sf,i}(X) = \frac{s(T)}{\gamma(X)} \left( \frac{\alpha(X)}{R_{sf,i}(SO_2)} + \frac{\beta(X)}{R_{sf,i}(O_3)} \right)^{-1} \qquad i \in \{v, s, g\} \tag{10}$$

where $R_{sf,i}(SO_2)$ and $R_{sf,i}(O_3)$ are tabulated resistances (Table S1) for each surface type, and $\alpha(X)$ and $\beta(X)$ are weighting factors (Table S2) estimated using the solubility (for $\alpha$) and reactivity (for $\beta$) of $X$ (Wesely, 1989; Zhang et al., 2002), $s(T)$ is a temperature adjustment factor (Zhang et al., 2003), and $\gamma(X)$ is a codeposition adjustment, which reflects changes in $R_{sf,i}(X)$ associated with surface acidity (Erisman et al., 1994; Massad et al., 2010; Neirynck et al., 2011; Wu et al., 2016). Here, we use the parameterizations of Massad et al. (2010) for $NH_3$ and Simpson et al. (2003) for $SO_2$:

$$\gamma(X) = \begin{cases} \exp(2 - r_{SN}) & X = SO_2 \text{ and } \alpha_{SN} \leq 2 \\ 6.35 r_{SN} & X = NH3 \\ 1 & \text{otherwise} \end{cases} \tag{11}$$

To avoid unrealistic oscillations in $v_d(NH3)$ and $v_d(SO_2)$, we estimate the acid ratio ($r_{SN}$) using the ratio of the 24-hour integrated total dry deposition of acids to the dry deposition of ammonia and ammonium, rather than using the ratio of their surface concentrations (Massad et al., 2010; Simpson et al., 2003).

The bidirectional exchange of ammonia is not represented in AM3-LM3-DD (Massad et al., 2010; Flechard et al., 2013). This reflects in part uncertainties in the emission potential of vegetation and the lack of detailed treatment of agricultural activities in LM3 (Riddick et al., 2016). We thus expect AM3-LM3-DD to overestimate $NH_3$ dry deposition in source regions (Zhu et al., 2015; Sutton et al., 2007).

## 2.2 Experimental design

We perform two sets of global simulations representative of present-day (circa 2010) and future (2050) conditions. For present-day conditions, AM3-LM3-DD is run from 2007 to 2010 using 2007 as spin-up. The model is forced with observed sea surface temperatures and sea ice cover, and land use from the Representative Concentration Pathways 8.5 scenario (RCP8.5, Riahi et al. (2011)). Anthropogenic emissions are from the Hemispheric Transport of Air Pollution 2 (HTAPv2, Janssens-Maenhout et al. (2015)). Natural emissions are based on Naik et al. (2013), except for isoprene emissions, which are calculated interactively using the Model of Emissions of Gases and Aerosols from Nature (MEGAN, Guenther et al. (2006)). This simulation will be referred to as R2010 hereafter.

An additional sensitivity experiment is performed (R2010_no_lu) in which natural vegetation is assumed to cover all vegetated tiles (i.e., no human land use). In both experiments, horizontal winds are nudged to those from the National Centers for Environmental Prediction reanalysis (Kalnay et al., 1996) to minimize meteorological variability between R2010 and R2010_no_lu.

For 2050, we use the vegetation, sea surface temperatures, and sea ice cover simulated by the GFDL-CM3 model under the RCP8.5 scenario in 2050 (Levy et al., 2013). RCP8.5 anthropogenic emissions for 2050 are used (Lamarque et al., 2011) except for $NH_3$, where we use the spatial distribution and seasonality of HTAPv2 emissions following Paulot et al. (2016). The model is run for 10 years with land-use fixed to year 2050 and we use the average of the last 9 years to minimize the impact of internal variability. This simulation will be referred to as R2050 hereafter. We perform two additional sensitivity experiments to characterize how land-use change (R2050_2010lu) and climate (R2050_2010climate) contribute to the change in deposition velocity between R2010 and R2050. The different model runs are summarized in Table 1.

**Table 1.** Model runs

| Run ID | Climate | Land Use | Anthropogenic Emissions |
|---|---|---|---|
| R2010 | 2008–2010[a] | RCP8.5 (2008–2010) | HTAPv2 |
| R2010_no_lu | 2008–2010[a] | natural vegetation | HTAPv2 |
| R2050 | 2050 | RCP8.5 (2050) | RCP8.5 (2050)[b] |
| R2050_2010lu | 2008–2010[a] | RCP8.5 (2008–2010) | RCP8.5 (2050)[b] |
| R2050_2010climate | 2008–2010[a] | RCP8.5 (2050) | RCP8.5 (2050)[b] |

[a] horizontal winds are nudged to NCEP

[b] with modified $NH_3$ emissions following Paulot et al. (2016)

## 3 Results and discussion

### 3.1 Evaluation of simulated $v_d$ against observations

The resistance approach for calculating dry deposition velocities implemented in AM3-LM3-DD is similar to that used in most chemical transport models. However differences in implementations can result in large differences between simulated deposition velocities (Wu et al., 2018). To illustrate these differences, Fig. 2 shows the sensitivity of $v_d(SO_2)$ and $v_d(NH_3)$ to temperature, wetness, and surface acidity in three global models: MOZART (Emmons et al., 2010), GEOS-Chem (Wang et al., 1998), and AM3-LM3–DD. Under dry conditions, GEOS-Chem and AM3-LM3–DD produce identical results for $v_d(SO_2)$, with the temperature dependence driven by that of the stomatal conductance. At low and high temperatures, $v_d(NH_3)$ is faster in AM3-LM3–DD than GEOS-Chem, which reflects small differences in the assumed surface pH (6.35 and 6.6 respectively). In contrast, MOZART assumes a surface pH=5 and accounts for changes in the effective solubility of $SO_2$ and

$NH_3$ with temperature, similar to Nguyen et al. (2015). The increase in solubility with decreasing temperature results in faster $v_d(X)$ at cold temperature in MOZART, while the lower pH increases $v_d(NH_3)$ and decreases $v_d(SO_2)$. The impact of surface wetness on $v_d(X)$ is only considered in MOZART and AM3-LM3 DD. In MOZART the presence of dew more than doubles $v_d(SO_2)$ but reduces $v_d(NH_3)$ below 25°C. In contrast, both $v_d(NH_3)$ and $v_d(SO_2)$ increase in AM3-LM3–DD when the canopy is wet, which is supported by observations (Erisman et al., 1994, 1999; Massad et al., 2010). AM3-LM3–DD also accounts for the modulation of $R_{sf,v}(SO_2)$ and $R_{sf,v}(NH_3)$ by the acidity of the surface. Our results suggest that when $\alpha_{SN} = 2$, i.e. when the deposition of acids is twice as large as the deposition of bases, the impact of codeposition can be greater than that of canopy wetness. Our comparison suggests that the implementation of the Wesely scheme in MOZART, AM3-LM3 DD, and GEOS-Chem produce similar $v_d(SO_2)$ and $v_d(NH_3)$ (within 50%) under dry conditions and for temperatures close to 20°C. However, differences in the sensitivity of $v_d(SO_2)$ and $v_d(NH_3)$ to environmental conditions (temperature, wetness, acidity) can result in large differences (>2). Such differences highlight the need for detailed evaluation of $v_d(X)$ across a wide range of conditions and chemical species (Wu et al., 2018).

### 3.1.1  $v_d(SO_2)$

We first evaluate the simulated present-day (R2010) $v_d(SO_2)$ against a compilation of field-based $v_d(SO_2)$ observations (Table S3). We sample the simulated monthly $v_d(SO_2)$ at the location of the measurements in the tile that best represents the type of vegetation reported in the observations. When observations are available, we further distinguish between day-time and night-time as well as wet and dry conditions. For day-time and night-time observations, we sample the model from 8am to 5pm and 10pm to 4am, respectively. For wet conditions, we sample the model when the canopy wetness is greater than 10%.

Fig. 3 shows observed and simulated $v_d(SO_2)$ grouped among the four types of vegetation simulated by LM3 (deciduous, coniferous, tropical, and grass).

Simulated deposition velocities generally fall within a factor of 2 of the observations, with better agreement during the day than at night, when the model is biased high. This uncertainty range is similar to that reported by Wu et al. (2018) in different dry deposition models. More specifically, AM3-LM3-DD qualitatively captures the range of deposition velocities over forested ecosystems, including the slower deposition of $SO_2$ in winter than in summer and under dry than under wet conditions in deciduous forests, and the fast removal of $SO_2$ over coniferous forests. However, the model fails to capture the elevated $v_d(SO_2)$ (>1 cm/s) reported by several studies over grasslands. This may reflect uncertainties in the representation of ammonia emissions (e.g., no sub-grid heterogeneity), which could result in an underestimate of $SO_2$-$NH_3$ co-deposition over crops or fertilized grasslands (Nemitz et al., 2001; Flechard et al., 2013).

### 3.1.2 $v_d(\mathrm{HNO3, HCN, H_2O_2, OrgN})$

Fig. 4 shows the observed deposition velocities for $\mathrm{HNO_3}$, a range of organic nitrates (ISOPN, MVKN, PROPNN) derived from isoprene photooxidation (Paulot et al., 2009), HCN, and $\mathrm{H_2O_2}$. We refer the reader to Nguyen et al. (2015) for information regarding the site and Caltech observations. We compare these observations with the simulated deposition velocities at this site decomposed into its stomatal, cuticle (wet and dry), stem, and ground components.

To facilitate the comparison between simulated and observed deposition velocities, we use meteorological fields (wind speed, temperature, precipitation, and downward radiation) from the Modern-Era Retrospective Analysis for Research and Applications (MERRA) (Rienecker et al., 2011) to drive a standalone version of LM3-DD. This provides a more accurate representation of the site conditions than using meteorological fields simulated by AM3.

The compounds measured by Nguyen et al. (2015) have different chemical properties, allowing us to evaluate the representation of different deposition pathways in AM3-LM3–DD. In particular, $\mathrm{HNO_3}$ and $\mathrm{H_2O_2}$ have negligible cuticular resistance ($R_{surf,v} \simeq 0$) (Nguyen et al., 2015), such that $v_d(X) \simeq [R_a + R_{b,v}(X)]^{-1}$ (ground deposition is negligible). Fig. 4 shows that LM3-DD captures both $v_d(\mathrm{H_2O_2})$ and $v_d(\mathrm{HNO_3})$ well, including the faster deposition of $\mathrm{H_2O_2}$ relative to $\mathrm{HNO_3}$, consistent with the dependence of $R_b$ on $1/D_X \propto \sqrt{MW(X)}$ (equation 5). In contrast, the low solubility and low reactivity at the leaf surface of HCN produces a large non stomatal resistance ($R_{sf,v} \ggg 1$ s/m, Nguyen et al. (2015)), such that $v_d(\mathrm{HCN}) \simeq R_s(\mathrm{HCN})^{-1}$. Comparison of observed and modeled $v_d(\mathrm{HCN})$ suggests that the Ball-Berry-Leuning model captures the stomatal conductance well at this site. Since $R_a$, $R_{b,v}$, and $R_s$ are well represented over the measurement period, we use observations of $v_d(\mathrm{ISOPN})$, $v_d(\mathrm{MVKN})$, and $v_d(\mathrm{PROPNN})$ at this site to estimate $\alpha$ and $\beta$ for these organic nitrates (Eq. 10). We find that ($\alpha = 7$, $\beta = 1$) provide a reasonable fit for all organic nitrates. These parameters imply that the deposition of isoprene-derived organic nitrates is primarily controlled by dry cuticles with small contributions of stomata and stems. We note that these parameters imply a much greater solubility and reactivity of organic nitrogen than in other models (e.g., $\alpha = 0$, $\beta = 0.5$ in AURAMS (Zhang et al., 2002)). While we use these parameters globally, such large differences warrant further investigations, as the deposition of organic nitrogen may account for over 25% of the overall N deposition but remains rarely measured (Jickells et al., 2013).

Finally, we note that the comparison against SOAS observations points to a significant high bias in simulated night-time deposition velocity. During this time period, the deposition is dominated by wet cuticles, which reflects the formation of dew in LM3. Since this bias is found for all species including those with little surface resistance ($\mathrm{H_2O_2}$ and $\mathrm{HNO_3}$), it is likely to be associated with an underestimate of the stability of the nocturnal boundary layer.

### 3.2 Impact of land heterogeneities on present-day N deposition

Fig. 5 shows the simulated dry deposition of oxidized N (dominated by $HNO_3$) and reduced N (dominated by $NH_3$) as well as the total N deposition (wet+dry) in North America. As noted in previous studies (Zhang et al., 2012; Lamarque et al., 2013), the overall pattern of N deposition mirrors the underlying distribution of $NH_3$ and NO emissions, with high deposition in the Northeast and greater contribution of reduced nitrogen to N deposition in the US Midwest and North Carolina than elsewhere in the Eastern US.

The grid-cell average dry deposition represents the area-weighted sum of the deposition fluxes to the tiles that comprise each grid cell. Fig. 5 (middle column) shows that N deposition over natural vegetation is generally greater than the grid-cell average, which is consistent with faster deposition velocities over forests relative to grasslands (Finkelstein (2001); Hicks (2006) and Fig. S1). Overall, the simulated total N deposition to natural ecosystems exceeds the grid-box average deposition by 10 to 30% over most of the Eastern and Central US. This enhancement is largest in regions where land-use change has caused a large decrease in vegetation height and LAI (e.g., in the US Midwest and Northeast, Fig. S2) and smallest in regions with little agricultural activity (e.g., most of Canada) or where managed vegetation differs little in height and LAI from natural vegetation (e.g., in the Western US, Fig. S2). Fig. 5 (middle column) also shows that the dry deposition of $NH_x$ exhibits a greater enhancement over natural vegetation than the dry deposition of $NO_y$, consistent with the greater sensitivity of $v_d(NH_3)$ than $v_d(HNO_3)$ to surface properties (Fig. S3). The enhancements of the dry deposition of $NH_x$ over natural vegetation is likely to be underestimated in AM3-LM3-DD as the surface bidirectional exchange of $NH_3$ tends to reduce its deposition in source regions.

Fig. 5 (right column) also shows that water bodies receive more reduced N but less oxidized N through dry deposition than the grid-box average. These differences can be attributed to the large effective solubility of $NH_3$ in freshwater, which results in lower $R_{sf,g}(NH_3)$ than over vegetated surfaces ($R_{sf,g}(HNO_3)$ is low over all surfaces). Our model suggests that $v_d(HNO_3)$ is generally slower over water bodies than over vegetated surfaces because of the lower roughness height of water bodies (see Fig. S3). The westward increase in the ratio of $NH_3$ to NO emissions thus results in water bodies receiving less N than the average grid cell in the Eastern US and Canada but more in the Central and Western US. .

### 3.3 Impact of anthropogenic land-use change on present-day N deposition

Fig. 6 shows the change in dry $NO_y$, dry $NH_x$, and total N deposition associated with anthropogenic land-use change, which is estimated by comparing R2010 and R2010_no_lu. We find that anthropogenic land-use change reduces dry $NO_y$, dry $NH_x$, and total N deposition over the contiguous US by 8%, 26%, and 6%, respectively. The reduction in N deposition associated with anthropogenic

land-use change is largest in the Central and Eastern US, where deforestation has caused a large reduction in LAI and vegetation height (Fig. S2).

While anthropogenic land use is estimated to reduce the overall N deposition in the contiguous US, we find that it tends to increase the surface concentration of reactive nitrogen species, which leads to greater N deposition on the remaining natural vegetation. Fig. 6 shows that land use has important implications for N deposition at national parks, which are best represented by natural vegetation tiles. For instance, we find that anthropogenic land-use change is associated with a 14% reduction in the overall N deposition in the region of Shenandoah National Park, but an increase of 9% on natural vegetation in the same grid box. The slower removal of N near source regions also facilitates N export to remote regions, such as Eastern Canada, where N deposition (primarily through wet deposition) increases by more than 10%. This suggests that anthropogenic land-use change in North America has contributed to the increase of N deposition to natural ecosystems both near source regions and in remote receptor regions.

## 3.4  Implications for future N deposition

Fig. 7 shows the simulated difference between N deposition in 2008–2010 (R2010) and 2050 (R2050). This difference reflects changes in anthropogenic emissions as well as changes in climate and land properties induced by climate and land-use change. Total N deposition is projected to increase by 9% over the contiguous US. Most of the increase is driven by greater deposition in the Midwest and Western US associated with higher $NH_3$ emissions (+40%). In contrast, N deposition is projected to decrease in the Eastern US following the decrease of NO emissions (-47%, mostly in the Eastern US).

We find a small increase (<10%) in the deposition velocity of $HNO_3$ over most of the US between R2010 and R2050 (Fig. S4). This is attributed to a reduction in the land fraction devoted to agriculture in RCP8.5 between 2010 and 2050 (Davies-Barnard et al., 2014), which results in taller vegetation and higher LAI. The impact of this change in land use between 2010 and 2050 is larger for $v_d(NH_3)$, which increases by more >10% over most of the Midwest and Eastern US. However, in the Eastern and Midwest US, this increase is more than compensated by a reduction in acid deposition, which results in an overall decrease of $v_d(NH_3)$ of 10 to 20% over most of the Eastern US. This highlights the need to better characterize the impact of the co-deposition of acids and ammonia on the removal of ammonia to improve projection of future N deposition.

Fig. 7 also shows that trends in N deposition simulated for all land types tend to be amplified over natural vegetation, because of the faster deposition velocities as discussed earlier. In contrast, water bodies are projected to experience an increase in N deposition over most of the US, including in regions which experience an overall decrease in N deposition. This contrast is driven by the faster removal of $NH_3$ over water relative to managed vegetation, which result in greater sensitivity to changes in the emissions of reduced N. The different responses of N deposition on natural tiles

and water tiles are important for projections of N deposition in national parks, where N deposition to both vegetation and water bodies is of concern. For instance, the changes in N deposition to natural vegetation from 2010 to 2050 at Voyageurs and Shenandoah national parks are 30% greater than simulated in the grid box where they are located, while N deposition to water bodies in the Shenandoah region is projected to increase by 16%, even though overall N deposition for the grid decreases by 18% in this region.

## 4 Conclusions

Our study highlights the importance of accounting for surface heterogeneities and anthropogenic land use in modulating the magnitude and trend of N deposition. Here, we leverage the tiled structure of the GFDL land model to efficiently represent the sub-grid scale heterogeneity of surface properties and their evolution in a changing climate. We have shown that the shift of N emissions from oxidized to reduced N in North America will exacerbate the sensitivity of N deposition to small-scale heterogeneities, which highlights the need to improve the representation of non-stomatal surface resistances ($R_{sf,v}$, $R_{sf,s}$, and $R_{sf,g}$) including their modulation by canopy wetness and acidity (Flechard et al., 2013; Wentworth et al., 2016; Wu et al., 2018).

Our approach is best suited to long time scales (decadal to centennial) and is complementary to ongoing efforts to improve the representation of present-day N deposition using a combination of high-resolution models and observations (Schwede and Lear, 2014). Future work will aim at coupling the representation of dry deposition presented here to the N cycle in the GFDL land model (Gerber et al., 2010), which will enable us to represent the bidirectional exchange of $NH_3$ (Nemitz et al., 2001; Flechard et al., 2013; Bash et al., 2013) and improve our understanding of the impact of N deposition on ecosystems and carbon cycling (Magnani et al., 2007; Janssens et al., 2010; Fleischer et al., 2013, 2015).

*Acknowledgement.* This study was supported by NOAA Climate Program Office's Atmospheric Chemistry, Carbon Cycle, and Climate program. Caltech observations and JDC were supported by NSF (Grant #AGS-1240604). We thank V. Naik, A. Fiore, and J. Schnell for helpful comments.

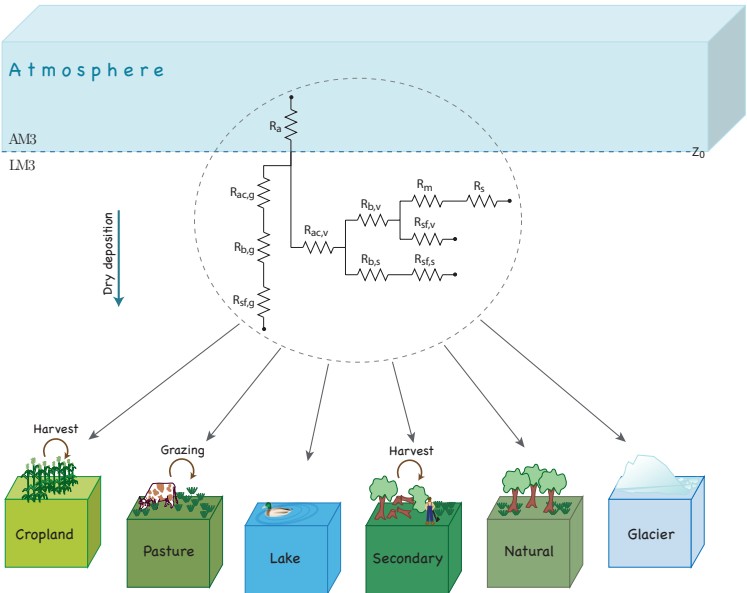

**Figure 1.** Schematic representation of the resistance scheme used to represent the dry deposition of gaseous tracers for each tile. $R_a$, $R_{b,i}$, $R_{ac}$, $R_m$ , $R_s$, and $R_{sf,i}$ are the aerodynamic resistance, laminar resistance, canopy aerodynamic resistance, mesophyll resistance, stomatal resistance, and surface resistance, respectively. The $g$, $s$, and $v$ indexes ($i$) refer to ground, stem, and vegetation. Note that for clarity deposition on soil and vegetation that are covered by snow or liquid water are not shown.

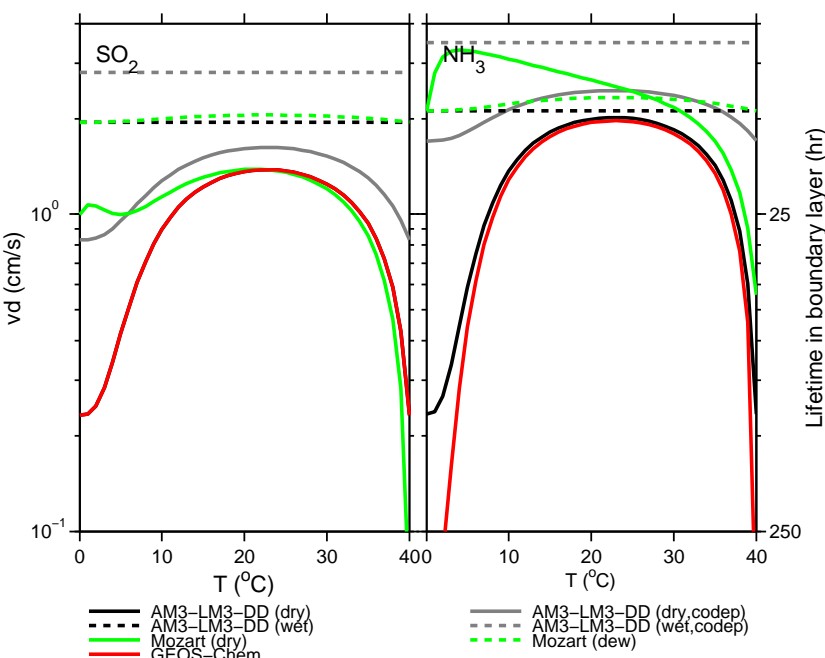

**Figure 2.** Simulated deposition velocity of $NH_3$ and $SO_2$ over a coniferous forest (LAI=5, $u_\star = 0.5$m/s, RH=80%) at different canopy wetness and temperature. To facilitate comparison across models, we use the same $R_a = 20$s/m, $R_b$ (Hicks et al., 1987) and $R_s$ (Wesely, 1989) for all models. Solar irradiation increases linearly from 0 to 800 W/m$^2$ with temperature. We neglect deposition to the ground and stems. Co-dep refers to the decrease in $R_{sf,v}(SO_2)$ and $R_{sf,v}(NH_3)$ associated with base and acid deposition respectively. For illustrative purposes, the ratio of acid to base deposition is set to 0.5 for $SO_2$ and 2 for $NH_3$. The lifetimes of $SO_2$ and $NH_3$ are estimated assuming a boundary layer height of 900m. GEOS-Chem and AM3-LM3–DD produce identical results for $SO_2$ under dry conditions.

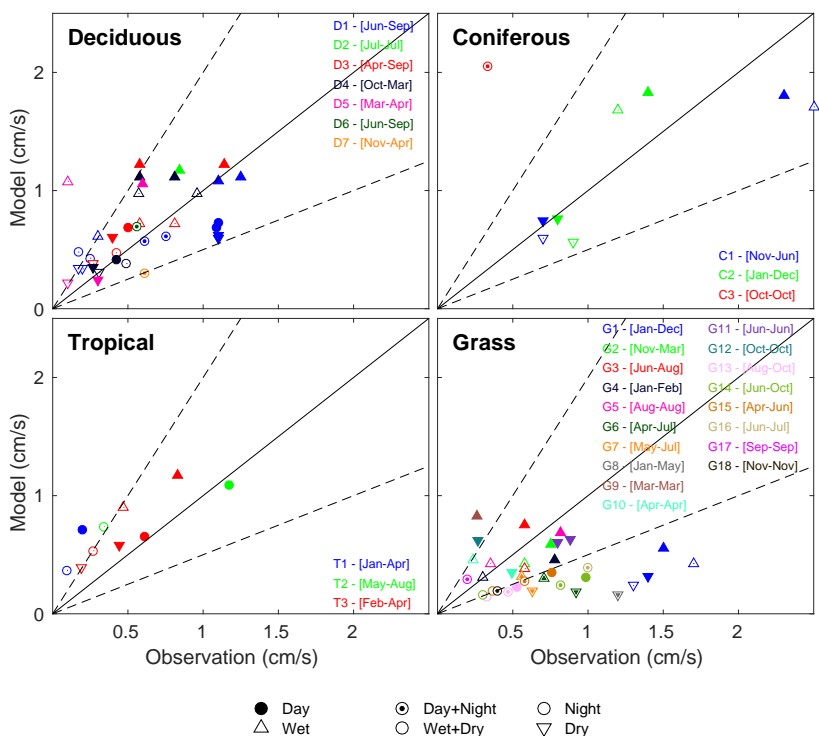

**Figure 3.** Observed and simulated deposition velocities of SO$_2$ for different vegetation types. The symbol shape indicates the canopy status: wet (upward pointing triangle), dry (downward point triangle), circle (average). The symbol fill indicates the time period: filled (day), half-filled (day+night), empty (night). The monthly diurnal cycle of deposition velocities simulated by AM3-LM3-DD (R2010 simulation) is sampled at each observation site in the tile that best represents the observed ecosystem accounting for the month, time of day and canopy wetness status when the observations were collected . References for the different sites are given in Table S3.

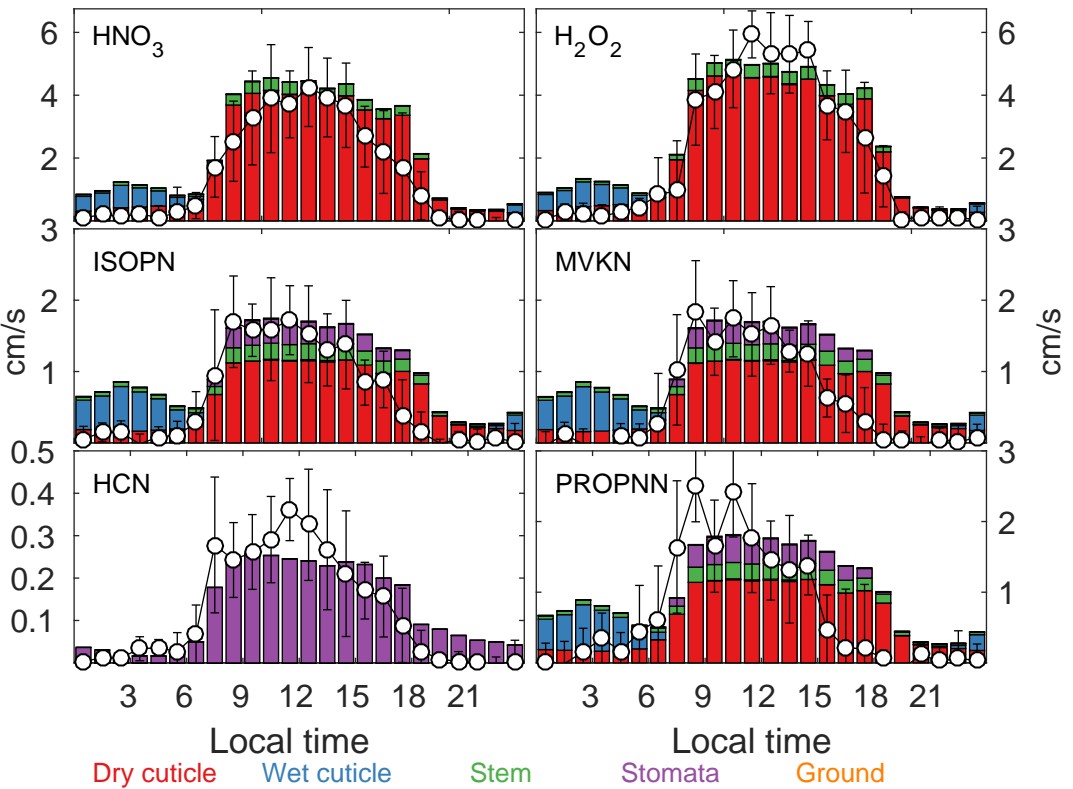

**Figure 4.** Observed (circles +/- standard deviation) and simulated (bars) dry deposition velocities for several nitrogen–containing species and hydrogen peroxide over the Talladega National Forest (Southeast US) in June 2013 (5 days, Nguyen et al. (2015)). The bar colors indicate the contribution of the different surfaces to the overall surface removal of the chemical tracer.

405

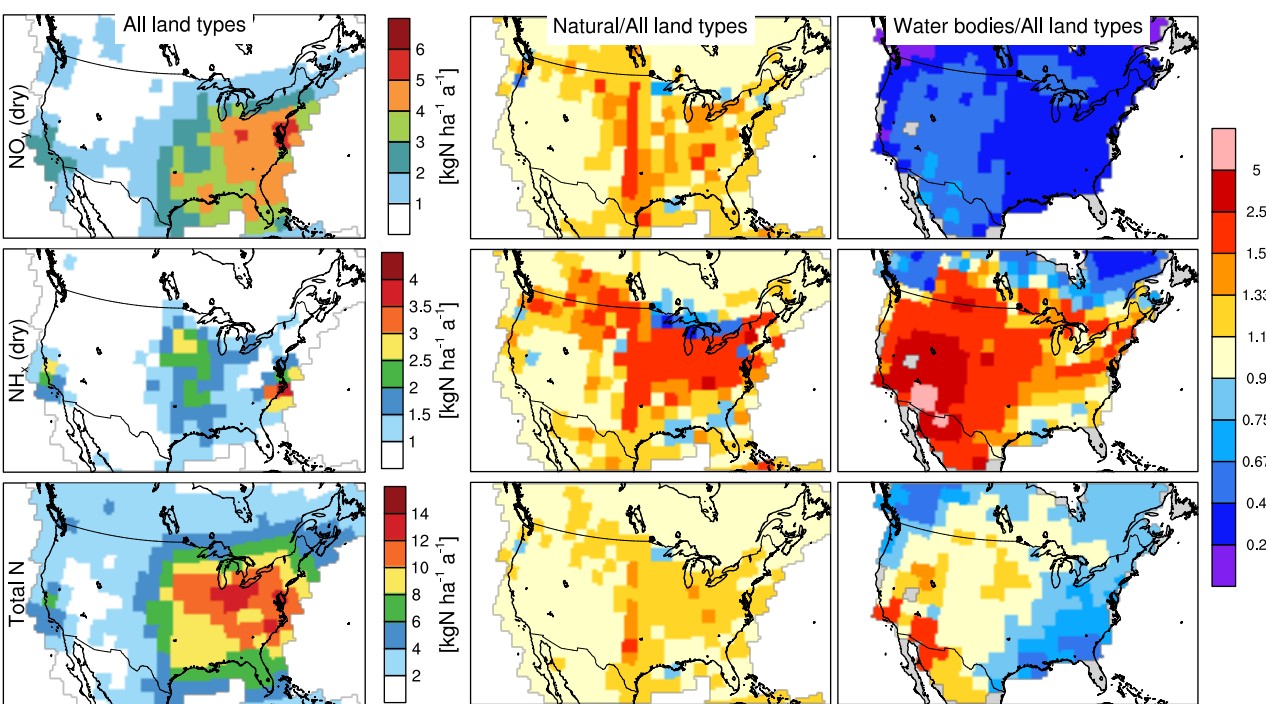

**Figure 5.** Simulated reactive nitrogen deposition (left column) from dry oxidized nitrogen deposition (first row), dry reduced nitrogen deposition (second row), and total nitrogen deposition (bottom row) over the 2008–2010 period. The ratio between the deposition on selected land types and the grid cell average deposition is shown in the middle (for natural vegetation) and right columns (water bodies)

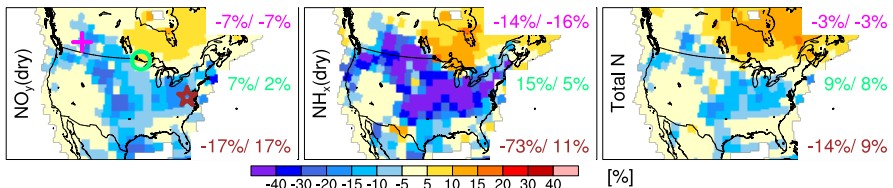

**Figure 6.** Relative change in the 2008–2010 average land deposition of dry oxidized nitrogen (left), dry reduced nitrogen (center), and total nitrogen (right) associated with anthropogenic land-use change. The relative change is shown as (with land-use - without land land-use)/with land-use. From top right to bottom right, the percentages indicate the change in N deposition at Banff National Park (cross), Voyageurs National Park (circle), and Shenandoah National Park (star) at the grid box level and on natural vegetation, a better proxy for these parks.

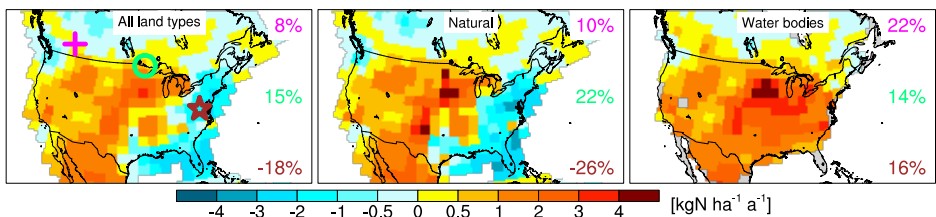

**Figure 7.** Simulated change in reactive nitrogen deposition from 2010 to 2050 in the RCP8.5 scenario at the grid box level (left), on natural tiles (center) and on water bodies (right). From top right to bottom right, the percentages indicate the change in N deposition at Banff National Park (cross), Voyageurs National Park (circle), and Shenandoah National Park (star) at the grid box level and on natural vegetation tiles respectively. The fractional change in N deposition over the contiguous US is indicated in inset (bottom left).

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
