# Peer review of "Representing sub-grid scale variations in nitrogen deposition associated with land use in a global Earth System Model: implications for present and future nitrogen deposition fluxes over North America"

_Atmospheric Chemistry and Physics, 2018_

## Referee Comment (RC1) · Anonymous Referee #1 · 4 Aug 2018

General Comments:

This paper describes a new modeling approach to allow for analysis of downscaled dry deposition values from an atmospheric-chemistry model that typically runs with a horizontal resolution of 200 km. Grid average values of dry deposition typically available from a coarse resolution model may not be relevant to ecological important processes that occur for specific land use types. This is an important contribution to the scientific

community and will be well used to support ecosystem assessments. The manuscript is a bit lacking in some important details and analyses. There should be some discussion early on in the paper about the bidirectional flux of NH3. It is now only mentioned as future work, but the lack of consideration of this in the current modeling has implications that should be discussed. With revision, the manuscript will be appropriate for publication in this journal.

Detailed Comments:

Line 44: Characterizing dry deposition as just surface resistance is not really accurate.

Line 46: The Schwede and Lear (2014) reference is not the appropriate one for the wet deposition fluxes as the values used are those from the National Atmospheric Deposition Program National Trends Network.

Line 55: The role of organic nitrogen should be discussed as well as other unmeasured components of the nitrogen budget that are currently only widely available from models.

Lines 72-79: Greater emphasis could be provided here that a new model has been developed for including in AM3. It isn't simply that you used an approach already in another model. You combined pieces from different models.

Line 80 – 93: The tile structure is a bit confusing and it isn't clear in the use of primary and secondary tiles and what information is contained at each level. What are the categories for the secondary tiles?

Line 86, I am not clear on the use of the phrase "transition rates". Maybe specifying them as temporal transitions would be helpful.

Line 93: LAI is a critical parameter for deposition. Please provide more details on how this is determined.

Line 96: Do the management practices influence the agricultural emissions as well?

Line 101: What is the basis for your assumption that 25% of the leaf biomass is removed daily by grazing?

Lines 102-103: Does LM3 include a tree growth model? How does changing the grazing frequency affect the growth of trees? Also, at line 103, the text is a bit garbled.

Lines 112-113: It would be helpful to provide the equations for Rac in this manuscript. The Bonan (1996) reference is not readily available and is not as commonly used at the Erisman approach.

Line 118: The leaf width is specified as one value for the land use type, while it is actually far more variable. How were the values in Table S1 determined? What is the sensitivity of the model to this parameter?

Line 119: Insert "species" before X here and other places.

Line 127: A right parenthesis is missing.

Line 129: After Rs(H2O), I suggest adding "is the stomatal resistance for water vapor and" before "is calculated"

Line 130-132: Water stress is included in most Jarvis based approaches which are commonly used in atmospheric chemistry models.

Line 137: The notation in Table S1 does not match the table in the manuscript. How are the scaling factors for stem/bark determined?

Lines 142-146: Please explain the nature of the modifications made to the original parameterization and how they were developed/evaluated.

Lines 148-165: The motivation for this section is not clear. Since you are only comparing between models, it is not an evaluation.

Section 2.2 – It would be helpful to include information about all model runs in this section. Later in the paper, several new runs are described that are not included here. It would also be helpful to include a table that summarizes the important options used

in the model runs. This could be included in the SI. What land use was used for the present day runs? Was there a spin-up year as was done for the future scenario run? What do you mean by across configurations at line 170? Were the emissions year specific? What year was used for the NH3 emissions that were used for the future scenario?

Section 2.3 (to be added) – There should be a section that describes the observational data used. How did you choose which ones to include? It might be helpful to include a table of the observational data to reduce the amount of text needed in the legend in Figure 3.

Lines 181-190: There is a lot more analysis that could be done in this section. For example, the differences in performance between land use types could be expanded. Are there aspects of the model that you think contribute to these differences? Canopy wetness is very important to SO2 deposition. How well does the model capture wetness compared to the observations?

Lines 191-213: This section mostly describes model development as the text describes how observations of deposition velocity were used to develop the alpha and beta parameters for equation 6. It isn't clear how the MERRA meteorological fields were included in the modeling. At line 200, reiterate that the measured compounds are those from Nguyen et al. At line 203, it would be appropriate to refer the reader back to Figure 4. How do you know what the deposition of HCN on cuticles is or are you referring to how the model treats the deposition?

Line 220: Note that the model captures the high reduced N over NC.

Line 222: In Figure 5 (middle column), what causes the streak pattern in the middle of the country?

Line 223-224: The text that appears at line 229-230 would be better placed here rather than simply referring vaguely to the supplementary materials.

Line 228: The land use is not actually changing in this analysis. Do you mean the the dry deposition of NHx would be more sensitive?

Line 235: What are you contrasting?

Line 240-241: This model run should be discussed in the methods section. How was this done in the model? Changes in the land use would also change the meteorology and the emissions. Were these considered?

Line 241: It would be helpful to insert text along the lines of "Using this run as the base case, we compare . . .". That would fit better with the text later in the section that compares the impact of anthropogenic land use changes.

Lines 254-256: Were changes in biogenic and agricultural emissions considered?

Line257: What land use takes the place of agricultural areas in the scenario?

Line 258-260: This section needs more explanation.

Line 270: Suggest adding "to natural vegetation from 2010-2050" after N deposition.

Line 273: Suggest adding "for the grid" after deposition.

Figure 3: What is the time scale for each point – e.g. monthly average? It isn't clear what you mean by "the model is sampled". The legend is far too small to be read easily. Perhaps some of the information could be included in a table. Some of the colors are hard to see or distinguish. The yellow doesn't show up at all in print and the blues are hard to distinguish. The symbols are small which makes it hard to tell them apart. Explain the symbol fill similar to how the shape is explained. How was the criteria for wet conditions determined? This could be explained in the methods section when you add a section for the observational data.

Figure 4: It would be helpful to have H2O2 and HNO3 on the same scale since these are compared in the manuscript. The surfaces listed are not consistent with the main text. Bark is listed rather than stem.

Figure 6: The wording of (with land-use – without land land-use)/with land-use is not clear.

Figure 7: Why include the 9% as an inset rather than simply stating it in the legend?

Figure S1: This figure has a lot of information in it and is not used as much as it could be. It seems like some of this would be important to discuss around line 115 in the main text, for example.

Figure S2: How do you determine the contribution of these factors?

Table S1: As noted above, the notation here does not match the main text.

---

## Referee Comment (RC2) · Anonymous Referee #3 · 14 Aug 2018

The paper deals with the coupling of a large-scale chemical transport model and a higher resolution land model to evaluate ecosystem level variations in reactive nitrogen deposition. The paper uses an interesting approach that could potentially also be applied to other global CTMs to evaluate land use specific variations. The conclusions of the paper have been shown in previous publications, see e.g. Simpson et al (2011) and references therein: http://www.nine-esf.org/files/ena_doc/ENA_pdfs/ENA_c14.pdf. In general, the paper is easy to follow, but there are some parts that need restructuring, or

additional explanation. This holds especially for the information on the measurements that have been used. I suggest that the authors address the following points before publication of the paper: General remarks: - The Introduction could be extended with a short overview of what has been done sofar regarding subgrid variations in deposition. - Could the authors add another short section about the observations that are used to validate the modelled Vds? The comparison between measurements and model results is difficult to interpret without this information. - Section 2.1: Figure 2 and its interpretation seems dislocated and should be part of the Result section. - Section 2.1: for NH3 the bi-directional flux using compensation point modelling is essential to model NH3 fluxes. Furthermore, for agricultural lands fertilization rates are important to determine the net flux of NH3. - The Experimental section does not include all the steps that are taken. It would help if the authors would explain in detail what their approach was.

- I would suggest to split up section 3.2. into two different sections: one section where Figure 5 is discussed and one that discusses the land use specific changes (Figure 6). I suggest to also split up section 3.3. One section that discusses the relative changes of anthropogenic land use changes on oxidized/reduced/total N deposition, and one about the contribution of natural land and water bodies to the total change in Nr deposition.

- Figure 4 shows clear overestimation during nighttime and at the end of the day. Could the authors discuss this, also in relation to the conclusions they draw from the comparison in Figure 4?

- Can the authors elaborate on the land use changes that are used in the model? What are for instance the regions where we see the largest changes?

- Could the authors elaborate a bit more on how the changes in N deposition in natural parks (mentioned at the end of section 3.3) are computed? Did the authors assume that the national parks cover an entire grid cell or did they for instance use a mask on

the level of the land model?

Detailed remarks: Page 3 line 80-82: this seems obvious. However, can the authors give examples where a comprehensive land model is included or at least a zooming option?

Page 3 line 88: can the authors explain a little more about the tiles sizes and its use in the mosaic approach, including information about the gridcell sizes?

Page 4 line 104: management practices are important, but that also holds for fertilization.

Page 4 line 109: 25% of biomass removed during grazing seems too high. Do the authors have a reference?

Page 4 line 112 – for crops, the representation of management practices needs some more explanation. Could you elaborate on how the planting and harvesting dates were determined?

Page 6 line 158 – I suggest to move the comparison of Vd from different models (the part related to Fig 2.) to the results and discussion section.

Page 6 Figure 3: it is not clear how the simulation was done: same locations? Actual meteorology or modelled? Surface characteristics?

Page 7 – line 192 – I suggest to change the title 'Evaluation' into something more specific (for instance 'Evaluation of model Vd against observations', or something on that line). The beginning of this sections should be moved to the experimental description.

Figure 5 – could the authors add the time period of the simulations to the description of the figure, so that it is self-explanatory. The titles 'All land' are a bit vague, maybe it is better to use 'All land types' or something in that direction. How many observations were used? Furthermore, can the authors explain the pattern in nitrogen deposition in central US, which is clear in the middle part (natural/all land)?

Page 8 line 237 – How would fertilization raters and the bi-directional nature of the NH3 flux influence the results in areas near to agricultural regions?

Figure 6 – all the numbers that are mentioned on the sides of the figure are a bit hard to follow. Please consider presenting the number in a different way.

Figure 7 – same as Figure 6.

[Figure]

---

## Referee Comment (RC3) · Anonymous Referee #2 · 24 Aug 2018

The authors addressed well my previous comments during the initial assessment of the earlier version of this paper, and this ACPD version has a better focus in terms of the study goals. This work will complementary to ongoing efforts to quantify nitrogen deposition at various spatial scales and provide an approach for assessing future nitrogen deposition scenarios from changing climate, land use, and emissions. I only have a few additional minor comments below for the authors to consider.

[Figure]

Based on the discussions in Section 3.1, I feel that the modelled Vd used in this study may be biased higher, or at least among the upper-end range of existing models such as those shown in Wu et al. (2018). Very high Vd values for some of the N species measured in Nguyen et al., 2015 are close to or even higher than the maximum possible Vd controlled by aerodynamic and sublayer resistances, and cannot be reproduced by the existing dry deposition models even after adjusting model parameters to the upper range of reasonable values. To avoid too much overestimation of dry deposition, these values are not recommended to be generalized to other regions before more measurement evidences become available. As for the present study, I understand the typical magnitude of uncertainties in dry deposition estimation is about a factor of 2 (Wu et al., 2018), and such uncertainties should be included in the discussion of modeled results and associated impacts on ecosystem health assessment. A brief discussion and recommendation related to this issue may be helpful to readers.

A related point to the above comment: I noticed that the other two reviewers both recommended using the bi-directional approach for NH3 deposition. I agree this approach is better in theory, but may not be practical in global models with the current limited knowledge of emission potentials in various land uses. I feel that using unidirectional depositional approach for NH3 is still acceptable if the chosen dry deposition model provides conservative Vd for NH3 (which compensates some of the bidirectional fluxes). Under such a condition, the NH3 deposition estimates would be valid for non-fertilized land use types and would represent the upper-end estimates for agricultural areas. This point can be made in the revised paper.

---

## Author Comment (AC1) · 3 Nov 2018

We wish to thank all three reviewers for their detailed comments and thoughtful suggestions.

**1 Reply to the comments of reviewer 1**

It appears that the comments from reviewer 1 refer to the first version submitted to ACPD and not to the version published online. As a result, the line numbers do not match the online version.

1. **There should be some discussion early on in the paper about the bidirectional flux of NH3. It is now only mentioned as future work, but the lack of consideration of this in the current modeling has implications that should be discussed.**

   We have added the following text to the method section:

   *The bidirectional exchange of ammonia is not represented in AM3-LM3-DD [Massad et al., 2010, Flechard et al., 2013]. This reflects in part uncertainties in the emission potential of vegetation and the lack of detailed treatment of agricultural activities in LM3 [Riddick et al., 2016]. We thus expect AM3-LM3-DD to overestimate $NH_3$ dry deposition in source regions [Zhu et al., 2015, Sutton et al., 2007].*

   As a result, we expect that the simulated enhancement of N deposition over natural tiles relative to the average deposition over all land tiles may be an underestimate. This has been clarified in the analysis of the results as follows:
   *The enhancements of the dry deposition of $NH_x$ over natural vegetation is likely to be underestimated in AM3-LM3-DD as the surface bidirectional exchange of $NH_3$ tends to reduce its deposition in source regions.*

2. **Line 44: Characterizing dry deposition as just surface resistance is not really accurate.**
   This has been removed.

3. **Line 46: The Schwede and Lear (2014) reference is not the appropriate one for the wet deposition fluxes as the values used are those from the National Atmospheric Deposition Program National Trends Network.**
   This section had been removed in the online version of the manuscript.

4. **Line 55: The role of organic nitrogen should be discussed as well as other unmeasured components of the nitrogen budget that are currently only widely available from models.**

   We have added the following text in the discussion of the SOAS data:

   *We find that ($\alpha = 7$, $\beta = 1$) provide a reasonable fit for all organic nitrates. These parameters imply that the deposition of isoprene-derived organic nitrates is primarily controlled by dry cuticles with small contributions of stomata and stems. We note that these parameters imply a much greater solubility and reactivity of organic nitrogen than in other models (e.g., $\alpha = 0$, $\beta = 0.5$ in AURAMS [Zhang et al., 2002]). While we use these parameters globally, such large differences warrant further investigations, as the deposition of organic nitrogen may account for over 25% of the overall N deposition but remains rarely measured [Jickells et al., 2013].*

5. **Lines 72-79: Greater emphasis could be provided here that a new model has been developed for including in AM3. It is not simply that you used an approach already in another model. You combined pieces from different models.**

   The text was revised as follows:
   *Here, we describe the development of a new model, in which dry deposition of gaseous and aerosol species are calculated within the dynamic vegetation model LM3 [Shevliakova et al., 2009, Milly et al., 2014]. The combined model will be referred to as AM3-LM3-DD hereafter.*

6. **Line 80-93: The tile structure is a bit confusing and it is not clear in the use of primary and secondary tiles and what information is contained at each level. What are the categories for the secondary tiles? AND Line 86, I am not clear on the use of the phrase "transition rates". Maybe specifying them as temporal transitions would be helpful.**

We have expanded the model description to clarify the tiling structure of LM3 and the representation of land-use change.

*In LM3, land surface heterogeneity is represented using a sub-grid mosaic of tiles [Shevliakova et al., 2009, Malyshev et al., 2015] as illustrated in Fig. 1. Each tile has distinct energy and moisture balances for a vegetation–snow–soil column, biophysical properties, and exchanges of radiant and turbulent fluxes with the overlying atmosphere. LM3 predicts physical, biogeochemical, and ecological characteristics for each sub-grid land surface tile from the top of the vegetation canopy to the bottom of the soil column including leaves and canopy temperature, canopy-air specific humidity, stomatal conductance, snow cover and depth, runoff, vertical distribution of soil moisture, ice, and temperature. The land-use history is prescribed from the Hurtt et al. [2011] reconstruction for each grid cell in terms of annual transition rates among four distinct land-use types: undisturbed (hereafter referred to as natural), crops, pastures, and secondary vegetation. Secondary vegetation is defined in LM3 as the vegetation recovering after land-uses and land-cover changes and not currently managed. This includes all abandoned agricultural land as well as the land where wood was harvested at least once in prior years. The model keeps track of different recovery states by creating a secondary vegetation tile every time a disturbance occurs and simulation the subsequent vegetation regrowth in the tile. To avoid unrestricted growth of the number of tiles, the number of secondary vegetation tiles is limited to 10 per grid cell in the configuration of LM3 used here. When more than 10 secondary vegetation titles exist in a grid cell, secondary vegetation tiles with similar properties are merged [Shevliakova et al., 2009], while preserving water, energy, and carbon balances. Land properties that affect the surface removal of chemical tracers, such as snow cover, canopy wetness, surface and canopy temperature, leaf area index (LAI), stomatal conductance, and vegetation height are all prognostic [Shevliakova et al., 2009].*

7. **Line 93: LAI is a critical parameter for deposition. Please provide more details on how this is determined.**
We have added the following text:

*Vegetation carbon is partitioned into five pools: leaves, fine roots, sapwood, heartwood, and labile storage. The model simulates changes in vegetation and soil carbon pools, as well as the carbon exchange among these pools and the atmosphere. The sizes of the pools are modified daily depending on the carbon uptake and according to a set of allocation rules. Additionally, the model simulates changes in the vegetation carbon pools due to phenological processes, natural mortality, and fire. LAI is determined by vegetation leaf biomass and specific leaf area, prescribed for each vegetation type described below.*

8. **Line 96: Do the management practices influence the agricultural emissions as well?**

No, agricultural emissions are taken directly from HTAPv2 or CMIP5 estimates. We have added the following the text:

*The impact of management practices on the timing and magnitude of agricultural emissions (e.g., Paulot et al. [2014]) is not accounted for in AM3-LM3-DD.*

9. **Line 101: What is the basis for your assumption that 25% of the leaf biomass is removed daily by grazing**

We assume a high grazing intensity to prevent excessive accumulation of the biomass on pastures and the consequent misclassification of vegetation types on pasture as forests. This high grazing intensity does not lead to excessive overgrazing on pastures since no grazing occurs once pasture LAI drops below a prescribed limit (LAI=2 in our simulations). As a result, the long-term average rate of grazing is to the rate of growth of leaf biomass for the pastures where LAI is higher than 2.

10. **Lines 102-103: Does LM3 include a tree growth model? How does changing the grazing frequency affect the growth of trees? Also, at line 103, the text is a bit garbled**

LM3 includes a fully prognostic dynamic vegetation model, and as such it does simulate the growth of vegetation, including trees. To clarify this point, we modified the text as follows:

*Each vegetated tile has a unique vegetation type (C3 grass, C4 grass, temperate deciduous, coniferous, or tropical vegetation), which is determined based on biogeographical rules that take into account environmental conditions as well as current state of the vegetation in each tile [Shevliakova et al., 2009].*

We also clarified the effects of the grazing on the growth of vegetation, and therefore on the vegetation types that the model simulates for pasture types:

*This higher grazing frequency and intensity prevent the excessive growth of vegetation biomass on pastures in the tropics and mid latitudes, a problem which was noted in previous versions of LM3 [Malyshev et al., 2015] leading to misclassification of pasture vegetation cover as forests.*

11. **Lines 112-113: It would be helpful to provide the equations for Rac in this manuscript. The Bonan (1996) reference is not readily available and is not as commonly used at the Erisman approach.**

    We have added the equation for both $R_{ac,v}$ and $R_{ac,g}$ to the text

    $$R_{ac,g} = \frac{u_\star}{14(LAI + SAI)h} \tag{1}$$

    $$R_{ac,v} = (LAI + SAI) \cdot g_b \text{ with } g_b = 0.01(1 - exp(-3/2)/3)\sqrt{V} \tag{2}$$

    where $SAI$, $h$, and V are the stem area index, the height of the vegetation, and the normalized wind at the top of the canopy, respectively.

12. **Line 118: The leaf width is specified as one value for the land use type, while it is actually far more variable. How were the values in Table S1 determined? What is the sensitivity of the model to this parameter?**
    The values are taken from [Petroff and Zhang, 2010] (reference given in Table S1). $R_{b,v}$ scales like $lw^{1/3}$, therefore only large differences in leaf width (e.g., between coniferous and decidious) will significant modify $R_{b,v}$.

13. **Line 119: Insert species X here and other places.**
    corrected

14. **Line 127: A right parenthesis is missing.**
    corrected

15. **Line 129: After Rs(H2O), I suggest adding "is the stomatal resistance for water vapor" before "s calculated"**
    corrected

16. **Line 130-132: Water stress is included in most Jarvis based approaches which are commonly used in atmospheric chemistry models.**
    we have removed this statement

17. **Line 137: The notation in Table S1 does not match the table in the manuscript. How are the scaling factors for stem/bark determined?**

    thank you for noting this inconsistency. The notation used in Table S1 has been revised to match that used in the text. We use the estimate of Padro et al. [1993] for $SO_2$. We assume that $R_{sf,s}(O_3)/R_{sf,s}(SO_2)$ is the same as $R_{sf,v}(O_3)/R_{sf,v}(SO_2)$ in pasture. This has been clarified in the notes associated with Table S1.

18. **Lines 142-146: Please explain the nature of the modifications made to the original parameterization and how they were developed/evaluated.**

    Both the parameterization of Massad et al. [2010] and Simpson et al. [2003] rely on the surface concentrations of ammonia and acids to estimate the acidity of the surface. We find that this can create unrealistic oscillations in $v_d(NH3)$ and $v_d(SO_2)$. This issue can be limited by using the 24h-integrated dry deposition fluxes instead. The text as been modified as follows:

*To improve numerical stability, we estimate the acid ratio ($r_{SN}$) using the ratio of the 24-hour integrated total dry deposition of acids to the dry deposition of ammonia and ammonium, rather than using the ratio of their surface concentrations [Massad et al., 2010, Simpson et al., 2003].*

19. **Lines 148-165: The motivation for this section is not clear. Since you are only comparing between models, it is not an evaluation.**
    This now serves as preamble to the evaluation section.

20. **Section 2.2 It would be helpful to include information about all model runs in this section. Later in the paper, several new runs are described that are not included here. It would also be helpful to include a table that summarizes the important options used in the model runs. This could be included in the SI. What land use was used for the present day runs? Was there a spin-up year as was done for the future scenario run? What do you mean by across configurations at line 170? Were the emissions year specific? What year was used for the NH3 emissions that were used for the future scenario?**

    We have given an ID to each simulation and we have also added a Table (Table 1) to summarize the different model configurations. Section 2.2 was rewritten as follows.

    *We perform two sets of global simulations representative of present-day (circa 2010) and future (2050) conditions. For present-day conditions, AM3-LM3-DD is run from 2007 to 2010 using 2007 as spin-up. The model is forced with observed sea surface temperatures and sea ice cover, and land use from the Representative Concentration Pathways 8.5 scenario (RCP8.5, Riahi et al. [2011]). Anthropogenic emissions are from the Hemispheric Transport of Air Pollution 2 (HTAPv2, Janssens-Maenhout et al. [2015]). Natural emissions are based on Naik et al. [2013], except for isoprene emissions, which are calculated interactively using the Model of Emissions of Gases and Aerosols from Nature (MEGAN, Guenther et al. [2006]). This simulation will be referred to as R2010 hereafter. An additional sensitivity experiment is performed (R2010_no_lu) with no anthropogenic land-use change, which is achieved by removing all vegetated tiles but the natural ones (expanding the area of the natural tiles). In both experiments, horizontal winds are nudged to those from the National Centers for Environmental Prediction reanalysis [Kalnay et al., 1996] to minimize meteorological variability between R2010 and R2010_no_lu.*

    *For 2050, we use the vegetation, sea surface temperatures, and sea ice cover simulated by the GFDL-CM3 model under the RCP8.5 scenario in 2050 [Levy et al., 2013]. RCP8.5 anthropogenic emissions for 2050 are used [Lamarque et al., 2011] except for $NH_3$, where we use the spatial distribution and seasonality of HTAPv2 emissions following Paulot et al. [2016]. The model is run for 10 years with land-use fixed to year 2050 and we use the average of the last 9 years to minimize the impact of internal variability. This simulation will be referred to as R2050 hereafter. We perform two additional sensitivity experiments to characterize how land-use change (R2050_2010lu) and climate (R2050_2010climate) contribute to the change in deposition velocity between R2010 and R2050. The different model runs are summarized in Table 1*

21. ***Section 2.3 (to be added) There should be a section that describes the observational data used. How did you choose which ones to include? It might be helpful to include a table of the observational data to reduce the amount of text needed in the legend in Figure 3.***

    We performed a litterature survey to identify observations of $v_d(SO_2)$ over a wide range of environments. We have added a table in the supplementary materials (Table S3), summarizing the observations. The text was revised as follows:

    *We first evaluate the simulated $v_d(SO_2)$ in AM3-LM3–DD for present-day (2007-2010) against observations collected over a wide range of surfaces (Table S3). We sample the simulated monthly diurnal cycle of $v_d(SO_2)$ at the location of the measurements in the tile that best represents the type of vegetation reported in the observations. We further distinguish between day-time (8am-5pm) and night-time (10pm-4am) samples and wet (wet fraction of the canopy greater than 10%) and dry periods.*

22. **Lines 181-190: There is a lot more analysis that could be done in this section. For example, the differences in performance between land use types could be expanded. Are there aspects of the model that you think contribute to these differences? Canopy wetness is very important**

**to SO2 deposition. How well does the model capture wetness compared to the observations?**

We find that AM3-LM3-DD falls within a factor of two of most observations. This is consistent with the uncertainty reported by Wu et al. [2018] for state of the art dry deposition models. We have expanded the analysis of the model biases as follows:

*Simulated deposition velocities generally fall within a factor of 2 of the observations, with better agreement during the day than at night, when the model is biased high. This uncertainty range is similar to the one reported by Wu et al. [2018] for a range of dry deposition models. More specifically, AM3-LM3-DD qualitatively captures the range of deposition velocities over forested ecosystems, including the slower deposition of $SO_2$ in winter than in summer and under dry than under wet conditions in deciduous forests and the fast removal of $SO_2$ over coniferous forests. However, the model fails to capture the elevated $v_d(SO_2)$ (>1 cm/s) reported by several studies over grassland. This may reflect uncertainties in the representation of ammonia emissions (e.g., no sub-grid heterogeneity), which could result in an underestimate of $SO_2$-$NH_3$ co-deposition over crops or fertilized grasslands [Nemitz et al., 2001, Flechard et al., 2013].*

We have also clarified how the canopy wetness is estimated. The following text was added to the method section:

*The fraction of the canopy covered by liquid water ($f_l$) and snow ($f_s$) are estimated from the intercepted canopy liquid water mass ($w_l$) and snow mass ($w_s$) following Bonan [1996]:*

$$f_i = \left( \frac{w_i}{W_{i,max}} \right)^{\frac{2}{3}} \qquad i \in \{l, s\} \tag{3}$$

*where $W_{l,max} = 0.02 kg\,m^{-2}$ and $W_{s,max} = 0.2 kg\,m^{-2}$ are the maximum liquid water and snow holding capacities, respectively. If both snow and liquid water are present simultaneously, water and snow are assumed to be distributed independently of each others.*

Fig. 2 shows that AM3-LM3-DD captures the qualitative impact of canopy wetness on $v_d(SO_2)$. However, we agree with the reviewer that the treatment of canopy wetness and its impact on deposition velocities remain important uncertainties in current models. This has been emphasized in the conclusion, as follows:

*Our study highlights the importance of accounting for surface heterogeneities and anthropogenic land use in modulating the magnitude and trend of N deposition. Here, we leverage the tiled structure of the GFDL land model to efficiently represent the subgrid scale heterogeneity of surface properties and their evolution in a changing climate. We have shown that the shift of N emissions from oxidized to reduced N in North America will exacerbate the sensitivity of N deposition to small-scale heterogeneities, which highlights the need to improve the representation of non-stomatal surface resistances ($R_{sf,v}$, $R_{sf,s}$, and $R_{sf,g}$) including their modulation by canopy wetness and acidity [Flechard et al., 2013, Wentworth et al., 2016, Wu et al., 2018].*

23. **Lines 191-213: This section mostly describes model development as the text describes how observations of deposition velocity were used to develop the alpha and beta parameters for equation 6. It is not clear how the MERRA meteorological fields were included in the modeling. At line 200, reiterate that the measured compounds are those from Nguyen et al. At line 203, it would be appropriate to refer the reader back to Figure 4. How do you know what the deposition of HCN on cuticles is or are you referring to how the model treats the deposition?**

We use MERRA meteorological fields to drive LM3 (instead of AM3 simulated fields), as it allows to better capture meteorological condition at the SOAS site. The text was modified as follows:

*To facilitate the comparison between simulated and observed deposition velocities, we use meteorological fields (wind speed, temperature, precipitation, and downward radiation) from the Modern-Era Retrospective Analysis*

*for Research and Applications (MERRA) [Rienecker et al., 2011] to drive a standalone version of LM3-DD. This provides a more accurate representation of the site conditions than using meteorological fields simulated by AM3.*

HCN is poorly soluble and does not exhibit significant reactivity at the leaf surface [Nguyen et al., 2015]. This was clarified as follows:

*In contrast, the low solubility and low reactivity at the leaf surface of HCN produces a large non stomatal resistance [Nguyen et al., 2015] ($R_{sf,v} >>> 1$ s/m), such that $v_d(\text{HCN}) \simeq R_s(\text{HCN})^{-1}$.*

24. **Line 220: Note that the model captures the high reduced N over NC.**

    The text has been modified as follows:

    *with high deposition in the Northeast and greater contribution of reduced nitrogen to N deposition in the US Midwest and North Carolina than in the Eastern US.*

25. ***Line 222: In Figure 5 (middle column), what causes the streak pattern in the middle of the country?***

    This reflects the large difference in vegetation height between the actual vegetation height and that of natural vegetation.

    The text was revised as follows:
    *This enhancement is largest in regions where land-use change has caused a large decrease in vegetation height and LAI (e.g., in the US Midwest and Northeast, Fig. S2) and smallest in regions with little agricultural activity (e.g., most of Canada) or where managed vegetation differs little in height and LAI from natural vegetation (e.g., in the Western US, Fig. S2).*

    Fig. S2 shows the difference in vegetation height and LAI associated with land-use change.

26. **Line 223-224: The text that appears at line 229-230 would be better placed here rather than simply referring vaguely to the supplementary materials.**

    We are now referring to Fig. S1

27. **Line 228: The land use is not actually changing in this analysis. Do you mean the the dry deposition of NHx would be more sensitive?**
    We have modified the text as follows:
    *… also shows that the dry deposition of $\text{NH}_x$ exhibits a greater enhancement over natural vegetation than the dry deposition of $\text{NO}_y$, consistent with the greater sensitivity of $v_d(\text{NH}_3)$ than $v_d(\text{HNO}_3)$ to surface properties (Fig. S3).*

28. **Line 235: What are you contrasting?**
    We have removed *In contrast*.

29. **Line 240-241: This model run should be discussed in the methods section. How was this done in the model? Changes in the land use would also change the meteorology and the emissions. Were these considered?**

    The effect of land-use on meteorology is limited by nudging the horizontal wind speeds to those from NCEP and prescribing the sea surface temperature and sea ice. This has been clarified in the method section (see reply to comment 20)

30. **Line 241: It would be helpful to insert text along the lines of "Using this run as the base case, we compare . . ." That would fit better with the text later in the section that compares the impact of anthropogenic land use changes.**

    The text was modified as follows:
    *Fig. 6 shows the change in dry $\text{NO}_y$, dry $\text{NH}_x$, and total N deposition associated with anthropogenic land-use change, which is estimated by comparing R2010 and R2010_no_lu. We find that anthropogenic land-use*

*change reduces dry* $NO_y$*, dry* $NH_x$*, and total N deposition over the contiguous US by 8%, 26%, and 6%, respectively. The reduction in N deposition associated with anthropogenic land-use change is largest in the Central and Eastern US, where deforestation has caused a large reduction in LAI and vegetation height (Fig. S4).*

31. **Lines 254-256: Were changes in biogenic and agricultural emissions considered?**

Emissions are either calculated (e.g., for isoprene) or read from monthly files on the atmospheric side and are not calculated in LM3. This has been clarified in the method section as described in reply to comment 8 from reviewer 1.

32. **Line 257: What land use takes the place of agricultural areas in the scenario?**

Only natural vegetation, glacier, and water tiles are considered in this simulation. This has been clarified in the revised experimental design (see reply to comment 20)

33. **Line 258-260: This section needs more explanation.**

We have revised the text as follows:

*We find a small increase (<10%) in the deposition velocity of* $HNO_3$ *over most of the US between R2050 and R2010 (Fig. S4). This is attributed to a reduction in the land fraction devoted to agriculture between 2010 and 2050 in the RCP8.5 [Davies-Barnard et al., 2014], which results in taller vegetation and higher LAI. The impact of this change in land use between 2010 and 2050 is larger for* $v_d(NH_3)$*, which increases by more >10% over most of the Midwest and Eastern US. However, in the Eastern US and US Midwest, this increase is more than compensated by a reduction in acid deposition, which results in an overall decrease of* $v_d(NH_3)$ *of 10 to 20% over most of the Eastern US. This highlights the need to better characterize the impact of the co-deposition of acids and ammonia on the removal of ammonia to improve projection of future N deposition.*

34. **Line 270: Suggest adding "to natural vegetation from 2010-2050" after N deposition.**
done

35. **Line 273: Suggest adding "for the grid" after deposition.**
done

36. **Figure 3: What is the time scale for each point – e.g. monthly average? It isn't clear what you mean by "the model is sampled". The legend is far too small to be read easily. Perhaps some of the information could be included in a table. Some of the colors are hard to see or distinguish. The yellow doesn't show up at all in print and the blues are hard to distinguish. The symbols are small which makes it hard to tell them apart. Explain the symbol fill similar to how the shape is explained. How was the criteria for wet conditions determined? This could be explained in the methods section when you add a section for the observational data.**

We have revised Fig. 4 following the comments of both reviewers 1 and 2 and expanded both the figure's caption and associated method section. See reply to comment 21.

37. **Figure 4: It would be helpful to have H2O2 and HNO3 on the same scale since these are compared in the manuscript. The surfaces listed are not consistent with the main text. Bark is listed rather than stem.**

we have revised the figure following the reviewer's suggestion.

**2 Reviewer 2**

1. **The Introduction could be extended with a short overview of what has been done sofar regarding subgrid variations in deposition.**

We have added the following text to the introduction:

*Significant challenges remain in quantifying the long-term impacts of N deposition on ecosystems in a changing climate [Sutton et al., 2008, Wu and Driscoll, 2010, Phoenix et al., 2012, Högberg, 2012, de Vries et al., 2015, Storkey et al., 2015], including uncertainties in the speciation, magnitude and spatial distribution of the N deposition flux itself [Sutton et al., 2008, Ochoa-Hueso et al., 2011, Jickells et al., 2013, Fleischer et al., 2013]. Many approaches have been developed to provide high-resolution, ecosystem-relevant estimates of both wet and dry N deposition, including statistical models [Singles et al., 1998, Dore et al., 2007, Weathers et al., 2006, Dore et al., 2012], high-resolution nested chemical transport model ($\simeq 4 \times 4\,km$ [Vieno et al., 2009, Simkin et al., 2016]), and hybrid approaches that combine high-resolution regional chemical transport models with observed N fluxes and atmospheric concentrations (e.g. using the Community Multiscale Air Quality Modeling System [Schwede and Lear, 2014, Bytnerowicz et al., 2015, Williams et al., 2017]). However, the elevated computational requirement associated with high-resolution atmospheric models make such approaches impractical for assessing the long-term impact of N deposition on ecosystems, its sensitivity to climate change, and ultimately its coupling with the carbon cycle [Smith et al., 2014, Zaehle et al., 2010, Fleischer et al., 2013, Dirnböck et al., 2017, Fleischer et al., 2015]. For such questions, estimates of N deposition are generally derived from global models with coarse resolution ($\simeq 100km$, [Dentener et al., 2006, Lamarque et al., 2013]). This introduces a large uncertainty [Hertel, 2011] in N deposition estimates especially for dry deposition, which can vary over short distances ($\sim 1\,km$) in response to changes in the physical, hydrological, and ecological state of the surface [Weathers et al., 2000, Hicks, 2006, 2008, De Schrijver et al., 2008, Ponette-González et al., 2010, Templer et al., 2014, Tulloss and Cadenasso, 2015].*

2. **Could the authors add another short section about the observations that are used to validate the modelled Vds? The comparison between measurements and model results is difficult to interpret without this information.**

We have split the model evaluation section into two sub sections. We have added a table summarizing the observation shown in Fig. 2 in the supplementary materials. See replies to comments 21 and 22 from reviewer 1.

3. **Section 2.1: Figure 2 and its interpretation seems dislocated and should be part of the Result section.**

This section was moved to the evaluation section.

4. **Section 2.1: for NH3 the bi-directional flux using compensation point modelling is essential to model NH3 fluxes. Furthermore, for agricultural lands fertilization rates are important to determine the net flux of NH3.**

The current representation of agriculture in LM3 is not sufficiently detailed to represent the bidirectional exchange. This has been clarified in the model description section in addition to the conclusion (see reply to comment 1 from reviewer 1).

We have further noted that the bidirectional nature of ammonia exchange should increase the enhancement of NHx deposition relative to the grid-box average. The text was modified as follows:

*The enhancements of the dry deposition of $NH_x$ over natural vegetation is likely to be underestimated by AM3-LM3-DD as the surface bidirectional exchange of $NH_3$ tends to reduce its deposition in source regions.*

5. **The Experimental section does not include all the steps that are taken. It would help if the authors would explain in detail what their approach was.**

We have revised the experimental section according to comments from both reviewers 1 and 2 (see reply to comment 20 from reviewer 1). We have also added a table to the main text (Table 1), which summarizes the different model configurations used in this study.

6. **I would suggest to split up section 3.2. into two different sections: one section where Figure 5 is discussed and one that discusses the land use specific changes (Figure 6). I suggest to also split up section 3.3. One section that discusses the relative changes of anthropogenic land use changes on oxidized/reduced/total N deposition, and one about the contribution of natural**

**land and water bodies to the total change in Nr deposition.**

Thank you. We have followed the reviewer's suggestion for section 3.2 that we have expanded. For section 3.3, we think it's clearer not to split the discussion of Fig. 7.

7. **Figure 4 shows clear overestimation during nighttime and at the end of the day. Could the authors discuss this, also in relation to the conclusions they draw from the comparison in Figure 4?**

The overestimate is found for all species but HCN. Because H2O2 and HNO3 have little surface resistance, we hypothesize that the overestimate is associated with insufficient aerodynamic resistance at night. The text is modified as follows:

*Finally, we note that the comparison against SOAS observations points to a significant high bias in simulated night-time deposition velocity. Since this bias is found for all species including $v_d(\mathrm{H_2O_2})$ and $v_d(\mathrm{HNO_3})$, this suggests that the model underestimates the aerodynamic resistance $(R_a)$ during these periods.*

8. **Can the authors elaborate on the land use changes that are used in the model? What are for instance the regions where we see the largest changes?**

We have added a figure in the supplementary materials that shows the change in LAI and vegetation height associated with land-use change (Fig. S2). We have also clarified how land-use change is implemented in LM3 in the method section as follows:

*The land-use history is prescribed from the Hurtt et al. [2011] reconstruction for each grid cell in terms of annual transition rates among four distinct land-use types: undisturbed (hereafter referred to as natural), crops, pastures, and secondary vegetation. Secondary vegetation is defined in LM3 as the vegetation recovering after land-uses and land-cover changes and not currently managed. This includes all abandoned agricultural land as well as the land where wood was harvested at least once in prior years. The model keeps track of different recovery states by creating a secondary vegetation tile every time a disturbance occurs and simulation the subsequent vegetation regrowth in the tile. To avoid unrestricted growth of the number of tiles, the number of secondary vegetation tiles is limited to 10 per grid cell in the configuration of LM3 used here. When more than 10 secondary vegetation titles exist in a grid cell, secondary vegetation tiles with similar properties are merged [Shevliakova et al., 2009], while preserving water, energy, and carbon balances.*

9. **Could the authors elaborate a bit more on how the changes in N deposition in natural parks (mentioned at the end of section 3.3) are computed? Did the authors assume that the national parks cover an entire grid cell or did they for instance use a mask on the level of the land model?**

We use N deposition to natural vegetation as a proxy for N deposition to natural parks. This has been clarified as follows:

*Fig. 5 shows that it has important implications for N deposition at national parks, which are best represented by natural vegetation tiles.*

10. **Detailed remarks: Page 3 line 80-82: this seems obvious. However, can the authors give examples where a comprehensive land model is included or at least a zooming option?**

We are not aware that other global climate models have used such zooming options. As noted in reply to comment 1, we have expanded the introduction to highlight ongoing efforts to improve estimates of present-day N deposition.

11. **Page 3 line 88: can the authors explain a little more about the tiles sizes and its use in the mosaic approach, including information about the gridcell sizes?**

Please see reply to comment 6 from reviewer 1. Fig. S1 also shows the tile size for natural and water bodies in LM3.

12. **Page 4 line 104: management practices are important, but that also holds for fertilization.**

We agree with the reviewer. Howevever LM3 does not yet include a detailed treatment of agricultural activities. In other words, the cropping schedule does not affect ammonia emissions, which are prescribed on the atmospheric side.

13. **Page 4 line 109: 25% of biomass removed during grazing seems too high. Do the authors have a reference?**

see reply to comment 9 from reviewer 1

14. **Page 4 line 112 – for crops, the representation of management practices needs some more explanation. Could you elaborate on how the planting and harvesting dates were determined?**

For planting and harvesting, we use a climatology [Portmann et al., 2010]. We have clarified that LM3 does not calculate cropping schedule (e.g., [Bondeau et al., 2007]), as follows:

*LM3 does not estimate the cropping schedule (e.g., Bondeau et al. [2007]) and we specify planting and harvesting dates from the global monthly irrigated and rainfed crop areas climatology [Portmann et al., 2010].*

15. **Page 6 line 158 – I suggest to move the comparison of Vd from different models (the part related to Fig 2.) to the results and discussion section.**

We have moved this section to the evaluation section

16. **Page 6 Figure 3: it is not clear how the simulation was done: same locations? Actual meteorology or modelled? Surface characteristics?**

We have clarified how the comparison was performed as follows:
*We first evaluate the simulated present-day (R2010) $v_d(SO_2)$ against observations collected over a wide range of surfaces (Table S3). We sample the simulated monthly diurnal cycle of $v_d(SO_2)$ at the location of the measurements in the tile that best represents the type of vegetation reported in the observations. We further distinguish between day-time (8am-5pm) and night-time (10pm-4am) samples and wet (wet fraction of the canopy greater than 10%) and dry periods.*

17. **Page 7 – line 192 – I suggest to change the title 'Evaluation' into something more specific (for instance 'Evaluation of model Vd against observations', or something on that line). The beginning of this sections should be moved to the experimental description.**

We have revised the title for this section as suggested by the reviewer. We have kept the description of the standalone LM3 configuration in this section as it is not used elsewhere.

18. **Figure 5 – could the authors add the time period of the simulations to the description of the figure, so that it is self-explanatory. The titles 'All land' are a bit vague, maybe it is better to use 'All land types' or something in that direction. How many observations were used? Furthermore, can the authors explain the pattern in nitrogen deposition in central US, which is clear in the middle part (natural/all land)?**

We have revised Fig.5 (and Fig. 7) as suggested by the reviewer. The large enhancement of N deposition in the central US over natural vegetation reflects the higher vegetation height of natural vegetation relative to the average vegetation height across all land types.

The text was revised as follows:
*This enhancement is largest in regions where land-use change has caused a large decrease in vegetation height and LAI (e.g., in the US Midwest and Northeast, Fig. S2) and smallest in regions with little agricultural activity (e.g., most of Canada) or where managed vegetation differs little in height and LAI from natural vegetation (e.g., in the Western US, Fig. S2).*

Fig. S2 shows the difference in vegetation height and LAI associated with land-use change.

19. **Page 8 line 237 – How would fertilization raters and the bi-directional nature of the NH3 flux influence the results in areas near to agricultural regions?**

We expect that the simulated enhancement of N deposition over natural tiles relative to the average deposition over all land tiles may be an underestimate. This has been clarified in the analysis of the results as follows:
*The enhancements of the dry deposition of $NH_x$ over natural vegetation is likely to be underestimated in AM3-LM3-DD as the surface bidirectional exchange of $NH_3$ tends to reduce its deposition in source regions.*

20. **Figure 6 – all the numbers that are mentioned on the sides of the figure are a bit hard to follow. Please consider presenting the number in a different way.**

We have made the text much larger. We have also moved the fractional change in N deposition over the contiguous US to the text.

21. **Figure 7 same as Figure 6.**

    see reply to previous comment

**3   Reviewer 3**

1. **Based on the discussions in Section 3.1, I feel that the modelled Vd used in this study may be biased higher, or at least among the upper-end range of existing models such as those shown in Wu et al. (2018). Very high Vd values for some of the N species measured in Nguyen et al., 2015 are close to or even higher than the maximum possible Vd controlled by aerodynamic and sublayer resistances, and cannot be reproduced by the existing dry deposition models even after adjusting model parameters to the upper range of reasonable values. To avoid too much overestimation of dry deposition, these values are not recommended to be generalized to other regions before more measurement evidences become available. As for the present study, I understand the typical magnitude of uncertainties in dry deposition estimation is about a factor of 2 (Wu et al., 2018), and such uncertainties should be included in the discussion of modeled results and associated impacts on ecosystem health assessment. A brief discussion and recommendation related to this issue may be helpful to readers.**

   We agree with the reviewer that more measurements are needed to better constrain the deposition velocity of organic nitrate. We have added the following text:

   *We find that ($\alpha = 7$, $\beta = 1$) provide a reasonable fit for all organic nitrates. These parameters imply that the deposition of isoprene-derived organic nitrates is primarily controlled by dry cuticles with small contributions of stomata and stems. We note that these parameters imply a much greater solubility and reactivity of organic nitrogen than in other models (e.g., $\alpha = 0$, $\beta = 0.5$ in AURAMS [Zhang et al., 2002]). While we use these parameters globally, such large differences warrant further investigations, as the deposition of organic nitrogen may account for over 25% of the overall N deposition but remains rarely measured [Jickells et al., 2013].*

   In the section *Evaluation of simulated* $v_d$ *against observations*, we are now referring to the study by Wu et al. [2018] as follows:

   *The resistance approach implemented in AM3-LM3-DD is similar to that used in most chemical transport models and has been evaluated extensively. However differences in implementations can result in large differences between simulated deposition velocities [Wu et al., 2018].*

   This has been further emphasized in the conclusion:
   *We have shown that the shift of N emissions from oxidized to reduced N in North America will exacerbate the sensitivity of N deposition to small-scale heterogeneities, which highlights the need to improve the representation of non-stomatal surface resistances ($R_{sf,v}$, $R_{sf,s}$, and $R_{sf,g}$) including their modulation by canopy wetness and acidity [Flechard et al., 2013, Wentworth et al., 2016, Wu et al., 2018].*

2. **A related point to the above comment: I noticed that the other two reviewers both recommended using the bi-directional approach for NH3 deposition. I agree this approach is better in theory, but may not be practical in global models with the current limited knowledge of emission potentials in various land uses. I feel that using unidirectional depositional approach for NH3 is still acceptable if the chosen dry deposition model provides conservative Vd for NH3 (which compensates some of the bidirectional fluxes). Under such a condition, the NH3 deposition estimates would be valid for non-fertilized land use types and would represent the upper-end estimates for agricultural areas. This point can be made in the revised paper.**

We agree with the reviewer that the bidirectional exchange remains challenging to incorporate in global climate models. In addition to uncertainties in the NH3 emission potential of different vegetation types, the representation agricultural activities in LM3 (and many other global dynamic vegetation models) remains insufficient to represent ammonia emissions.

We have added the following text to the method section:

*The bidirectional exchange of ammonia is not represented in AM3-LM3-DD [Massad et al., 2010, Flechard et al., 2013]. This reflects in part uncertainties in the emission potential of vegetation and the lack of detailed treatment of agricultural activities in LM3 [Riddick et al., 2016]. We thus expect AM3-LM3-DD to overestimate* $NH_3$ *dry deposition in source regions [Zhu et al., 2015, Sutton et al., 2007].*

Table 1: Model runs

| Run ID | Climate | Land Use | Anthropogenic Emissions |
|---|---|---|---|
| R2010 | 2008–2010[a] | RCP8.5 (2008–2010) | HTAPv2 |
| R2010_no_lu | 2008–2010[a] | natural vegetation | HTAPv2 |
| R2050 | 2050 | RCP8.5 (2050) | RCP8.5 (2050)[b] |
| R2050_2010lu | 2008–2010[a] | RCP8.5 (2008–2010) | RCP8.5 (2050)[b] |
| R2050_2010climate | 2008–2010[a] | RCP8.5 (2050) | RCP8.5 (2050)[b] |

[a] horizontal winds are nudged to NCEP

[b] with modified $NH_3$ emissions following Paulot et al. [2016]

[Figure]

Figure S2: Overall change in LAI and vegetation height associated with anthropogenic land-use change (2008–2010 average)

[Figure]

Figure 4: Observed and simulated deposition velocities of SO$_2$ for different vegetation types. The symbol shape indicates the canopy status: wet (upward pointing triangle), dry (downward point triangle), circle (average). The symbol fill indicates the time period: filled (day), half-filled (day+night), empty (night). The monthly diurnal cycle of deposition velocities simulated by AM3-LM3-DD (R2010 simulation) is sampled at each observation site in the tile that best represents the observed ecosystem accounting for the month, time of day and canopy wetness status when the observations were collected . References for the different sites are given in Table S3.

**References**

Gordon B Bonan. Land surface model (lsm version 1.0) for ecological, hydrological, and atmospheric studies: Technical description and users guide. technical note. Technical report, National Center for Atmospheric Research, Boulder, CO (United States). Climate and Global Dynamics Div., 1996.

A. Bondeau, P.C. Smith, S. Zaehle, S. Schaphoff, W. Lucht, W. Cramer, D. Gerten, H. Lotze-Campen, C. Müller, M. Reichstein, et al. Modelling the role of agriculture for the 20th century global terrestrial carbon balance. *Global Change Biol.*, 13(3):679–706, 2007.

A. Bytnerowicz, R.F. Johnson, L. Zhang, G.D. Jenerette, M.E. Fenn, S.L. Schilling, and I. Gonzalez-Fernandez. An empirical inferential method of estimating nitrogen deposition to mediterranean-type ecosystems: the san bernardino mountains case study. *Environmental Pollution*, 203:69–88, aug 2015. doi: 10.1016/j.envpol.2015.03.028. URL https://doi.org/10.1016/j.envpol.2015.03.028.

T. Davies-Barnard, P. J. Valdes, J. S. Singarayer, F. M. Pacifico, and C. D. Jones. Full effects of land use change in the representative concentration pathways. *Environ. Res. Lett.*, 9(11):114014, 2014. ISSN 1748-9326. doi: 10.1088/1748-9326/9/11/114014. URL http://stacks.iop.org/1748-9326/9/i=11/a=114014.

A. De Schrijver, J. Staelens, K. Wuyts, G. Van Hoydonck, N. Janssen, J. Mertens, L. Gielis, G. Geudens, L. Augusto, and K. Verheyen. Effect of vegetation type on throughfall deposition and seepage flux. *Environ. Pollut.*, 153(2): 295–303, May 2008. ISSN 0269-7491.

Wim de Vries, Jean-Paul Hettelingh, and Maximilian Posch, editors. *Critical Loads and Dynamic Risk Assessments*, volume 25 of *Environmental Pollution*. Springer Netherlands, Dordrecht, 2015. ISBN 978-94-017-9507-4 978-94-017-9508-1.

F. Dentener, J. Drevet, J. F. Lamarque, I. Bey, B. Eickhout, A. M. Fiore, D. Hauglustaine, L. W. Horowitz, M. Krol, U. C. Kulshrestha, M. Lawrence, C. Galy-Lacaux, S. Rast, D. Shindell, D. Stevenson, T. Van Noije, C. Atherton, N. Bell, D. Bergman, T. Butler, J. Cofala, B. Collins, R. Doherty, K. Ellingsen, J. Galloway, M. Gauss, V. Montanaro, J. F. Müller, G. Pitari, J. Rodriguez, M. Sanderson, F. Solmon, S. Strahan, M. Schultz, K. Sudo, S. Szopa, and O. Wild. Nitrogen and sulfur deposition on regional and global scales: A multimodel evaluation. *Global Biogeochem. Cycles*, 20:B4003, October 2006.

T. Dirnböck, C. Foldal, I. Djukic, J. Kobler, E. Haas, R. Kiese, and B. Kitzler. Historic nitrogen deposition determines future climate change effects on nitrogen retention in temperate forests. *Climatic Change*, 144(2): 221–235, jul 2017. doi: 10.1007/s10584-017-2024-y. URL https://doi.org/10.1007/s10584-017-2024-y.

A Dore, M Vieno, Y Tang, U Dragosits, A Dosio, K Weston, and M Sutton. Modelling the atmospheric transport and deposition of sulphur and nitrogen over the united kingdom and assessment of the influence of SO2 emissions from international shipping. *Atmospheric Environment*, 41(11):2355–2367, apr 2007. doi: 10.1016/j.atmosenv.2006.11.013. URL https://doi.org/10.1016/j.atmosenv.2006.11.013.

A. J. Dore, M. Kryza, J. R. Hall, S. Hallsworth, V. J. D. Keller, M. Vieno, and M. A. Sutton. The influence of model grid resolution on estimation of national scale nitrogen deposition and exceedance of critical loads. *Biogeosciences*, 9(5):1597–1609, may 2012. doi: 10.5194/bg-9-1597-2012. URL https://doi.org/10.5194/bg-9-1597-2012.

C. R. Flechard, R.-S. Massad, B. Loubet, E. Personne, D. Simpson, J. O. Bash, E. J. Cooter, E. Nemitz, and M. A. Sutton. Advances in understanding, models and parameterizations of biosphere-atmosphere ammonia exchange. *Biogeosciences*, 10(7):5183–5225, July 2013. ISSN 1726-4189.

K. Fleischer, K. T. Rebel, M. K. van der Molen, J. W. Erisman, M. J. Wassen, E. E. van Loon, L. Montagnani, C. M. Gough, M. Herbst, I. A. Janssens, D. Gianelle, and A. J. Dolman. The contribution of nitrogen deposition to the photosynthetic capacity of forests. *Global Biogeochem. Cycles*, 27(1):187–199, March 2013. ISSN 1944-9224.

K. Fleischer, D. Wårlind, M. K. van der Molen, K. T. Rebel, A. Arneth, J. W. Erisman, M. J. Wassen, B. Smith, C. M. Gough, H. A. Margolis, A. Cescatti, L. Montagnani, A. Arain, and A. J. Dolman. Low historical nitrogen deposition effect on carbon sequestration in the boreal zone. *J. Geophys. Res. Biogeosci.*, 120(12):2015JG002988, December 2015. ISSN 2169-8961.

A. Guenther, T. Karl, P. Harley, C. Wiedinmyer, P. I. Palmer, and C. Geron. Estimates of global terrestrial isoprene emissions using MEGAN (Model of Emissions of Gases and Aerosols from Nature). *Atmos. Chem. Phys.*, 6(11): 3181–3210, 2006.

Ole Hertel. *The European Nitrogen Assessment: Sources, Effects and Policy Perspectives*, chapter 14. Cambridge University Press, May 2011. ISBN 1107006120.

Bruce B. Hicks. Dry deposition to forests—On the use of data from clearings. *Agric. For. Meteorol.*, 136(3–4):214–221, February 2006. ISSN 0168-1923. doi: 10.1016/j.agrformet.2004.06.013. URL http://www.sciencedirect.com/science/article/pii/S0168192305002066.

Bruce B. Hicks. On Estimating Dry Deposition Rates in Complex Terrain. *Journal of Applied Meteorology and Climatology*, 47(6):1651–1658, June 2008. ISSN 1558-8424.

Peter Högberg. What is the quantitative relation between nitrogen deposition and forest carbon sequestration? *Glob. Chang. Biol.*, 18(1):1–2, January 2012. ISSN 1365-2486.

G. C. Hurtt, L. P. Chini, S. Frolking, R. A. Betts, J. Feddema, G. Fischer, J. P. Fisk, K. Hibbard, R. A. Houghton, A. Janetos, C. D. Jones, G. Kindermann, T. Kinoshita, Kees Klein Goldewijk, K. Riahi, E. Shevliakova, S. Smith, E. Stehfest, A. Thomson, P. Thornton, D. P. van Vuuren, and Y. P. Wang. Harmonization of land-use scenarios for the period 1500–2100: 600 years of global gridded annual land-use transitions, wood harvest, and resulting secondary lands. *Clim. Change*, 109(1-2):117–161, November 2011. ISSN 0165-0009, 1573-1480.

G. Janssens-Maenhout, M. Crippa, D. Guizzardi, F. Dentener, M. Muntean, G. Pouliot, T. Keating, Q. Zhang, J. Kurokawa, R. Wankmüller, H. Denier van der Gon, J. J. P. Kuenen, Z. Klimont, G. Frost, S. Darras, B. Koffi, and M. Li. HTAP_v2.2: a mosaic of regional and global emission grid maps for 2008 and 2010 to study hemispheric transport of air pollution. *Atmos. Chem. Phys.*, 15(19):11411–11432, October 2015. ISSN 1680-7324. doi: 10.5194/acp-15-11411-2015. URL http://www.atmos-chem-phys.net/15/11411/2015/.

T. Jickells, A. R. Baker, J. N. Cape, S. E. Cornell, and E. Nemitz. The cycling of organic nitrogen through the atmosphere. *Philos. Trans. R. Soc. London, Ser. B*, 368(1621), July 2013. ISSN 0962-8436, 1471-2970. PMID: 23713115.

E. Kalnay, M. Kanamitsu, R. Kistler, W. Collins, D. Deaven, L. Gandin, M. Iredell, S. Saha, G. White, J. Woollen, Y. Zhu, A. Leetmaa, R. Reynolds, M. Chelliah, W. Ebisuzaki, W. Higgins, J. Janowiak, K. C. Mo, C. Ropelewski, J. Wang, Roy Jenne, and Dennis Joseph. The NCEP/NCAR 40-Year Reanalysis Project. *Bull. Am. Meteorol. Soc.*, 77(3):437–471, March 1996. ISSN 0003-0007.

J.-F. Lamarque, F. Dentener, J. McConnell, C.-U. Ro, M. Shaw, R. Vet, D. Bergmann, P. Cameron-Smith, S. Dalsoren, R. Doherty, G. Faluvegi, S. J. Ghan, B. Josse, Y. H. Lee, I. A. MacKenzie, D. Plummer, D. T. Shindell, R. B. Skeie, D. S. Stevenson, S. Strode, G. Zeng, M. Curran, D. Dahl-Jensen, S. Das, D. Fritzsche, and M. Nolan. Multi-model mean nitrogen and sulfur deposition from the atmospheric chemistry and climate model intercomparison project (ACCMIP): evaluation of historical and projected future changes. *Atmos. Chem. Phys.*, 13(16): 7997–8018, August 2013. ISSN 1680-7324.

Jean-François Lamarque, G. Kyle, Malte Meinshausen, Keywan Riahi, Steven Smith, Detlef van Vuuren, Andrew Conley, and Francis Vitt. Global and regional evolution of short-lived radiatively-active gases and aerosols in the representative concentration pathways. *Clim. Change*, 109:191–212, 2011. ISSN 0165-0009. 10.1007/s10584-011-0155-0.

Hiram Levy, Larry W. Horowitz, M. Daniel Schwarzkopf, Yi Ming, Jean-Christophe Golaz, Vaishali Naik, and V. Ramaswamy. The roles of aerosol direct and indirect effects in past and future climate change. *J. Geophys. Res. Atmos.*, 118(10):4521–4532, 2013. ISSN 2169-8996.

Sergey Malyshev, Elena Shevliakova, Ronald J Stouffer, and Stephen W Pacala. Contrasting local versus regional effects of Land-Use-Change-Induced heterogeneity on historical climate: Analysis with the GFDL earth system model. *J. Clim.*, 28(13):5448–5469, 2015.

R.-S. Massad, E. Nemitz, and M. A. Sutton. Review and parameterisation of bi-directional ammonia exchange between vegetation and the atmosphere. *Atmos. Chem. Phys.*, 10(21):10359–10386, 2010.

P. C. D. Milly, Sergey L. Malyshev, Elena Shevliakova, Krista A. Dunne, Kirsten L. Findell, Tom Gleeson, Zhi Liang, Peter Phillipps, Ronald J. Stouffer, and Sean Swenson. An enhanced model of land water and energy for global hydrologic and earth-system studies. *J. Hydrometeorol.*, 15(5):1739–1761, June 2014. ISSN 1525-755X.

Vaishali Naik, Larry W. Horowitz, Arlene M. Fiore, Paul Ginoux, Jingqiu Mao, Adetutu M. Aghedo, and Hiram Levy. Impact of preindustrial to present-day changes in short-lived pollutant emissions on atmospheric composition and climate forcing. *J. Geophys. Res. Atmos.*, 118(14):8086–8110, July 2013. ISSN 2169-8996.

Eiko Nemitz, Celia Milford, and Mark A. Sutton. A two-layer canopy compensation point model for describing bi-directional biosphere-atmosphere exchange of ammonia. *Quart. J. Roy. Meteor. Soc.*, 127(573):815–833, April 2001. ISSN 1477-870X.

Tran B. Nguyen, John D. Crounse, Alex P. Teng, Jason M. St Clair, Fabien Paulot, Glenn M. Wolfe, and Paul O. Wennberg. Rapid deposition of oxidized biogenic compounds to a temperate forest. *Proc. Natl. Acad. Sci. U.S.A.*, 112(5):E392–E401, February 2015. ISSN 0027-8424, 1091-6490. doi: 10.1073/pnas.1418702112. URL http://www.pnas.org/content/112/5/E392.

Raùl Ochoa-Hueso, Edith B. Allen, Cristina Branquinho, Cristina Cruz, Teresa Dias, Mark E. Fen, Esteban Manrique, M. Esther Pérez-Corona, Lucy J. Sheppard, and William D. Stock. Nitrogen deposition effects on mediterranean-type ecosystems: An ecological assessment. *Environ. Pollut.*, 159(10):2265 – 2279, 2011. ISSN 0269-7491.

J. Padro, H. H. Neumann, and G. Den Hartog. Dry deposition velocity estimates of $SO_2$ from models and measurements over a deciduous forest in winter. *Water Air Soil Pollut.*, 68(3-4):325–339, June 1993. ISSN 0049-6979, 1573-2932.

F. Paulot, D. J. Jacob, R. W. Pinder, J. O. Bash, K. Travis, and D. K. Henze. Ammonia emissions in the United States, European Union, and China derived by high-resolution inversion of ammonium wet deposition data: Interpretation with a new agricultural emissions inventory (MASAGE_NH3). *J. Geophys. Res. Atmos.*, 119(7): 4343–4364, April 2014. ISSN 2169-8996.

F. Paulot, P. Ginoux, W. F. Cooke, L. J. Donner, S. Fan, M.-Y. Lin, J. Mao, V. Naik, and L. W. Horowitz. Sensitivity of nitrate aerosols to ammonia emissions and to nitrate chemistry: implications for present and future nitrate optical depth. *Atmos. Chem. Phys.*, 16(3):1459–1477, 2016. doi: 10.5194/acp-16-1459-2016. URL http://www.atmos-chem-phys.net/16/1459/2016/.

A. Petroff and L. Zhang. Development and validation of a size-resolved particle dry deposition scheme for application in aerosol transport models. *Geosci. Model Dev.*, 3(2):753–769, December 2010. ISSN 1991-9603.

Gareth K. Phoenix, Bridget A. Emmett, Andrea J. Britton, Simon J. M. Caporn, Nancy B. Dise, Rachel Helliwell, Laurence Jones, Jonathan R. Leake, Ian D. Leith, Lucy J. Sheppard, Alwyn Sowerby, Michael G. Pilkington, Edwin C. Rowe, Mike R. Ashmore, and Sally A. Power. Impacts of atmospheric nitrogen deposition: responses of multiple plant and soil parameters across contrasting ecosystems in long-term field experiments. *Glob. Chang. Biol.*, 18(4):1197–1215, April 2012. ISSN 1365-2486.

A. G. Ponette-González, K. C. Weathers, and L. M. Curran. Tropical land-cover change alters biogeochemical inputs to ecosystems in a Mexican montane landscape. *Ecol. Appl.*, 20(7):1820–1837, January 2010. ISSN 1939-5582.

Felix T. Portmann, Stefan Siebert, and Petra Döll. Mirca2000 — global monthly irrigated and rainfed crop areas around the year 2000: A new high-resolution data set for agricultural and hydrological modeling. *Global Biogeochem. Cycles*, 24(1):GB1011, 03 2010.

Keywan Riahi, Shilpa Rao, Volker Krey, Cheolhung Cho, Vadim Chirkov, Guenther Fischer, Georg Kindermann, Nebojsa Nakicenovic, and Peter Rafaj. Rcp 8.5—a scenario of comparatively high greenhouse gas emissions. *Clim. Chang.*, 109(1-2):33–57, 2011.

Stuart Riddick, Daniel Ward, Peter Hess, Natalie Mahowald, Raia Massad, and Elisabeth Holland. Estimate of changes in agricultural terrestrial nitrogen pathways and ammonia emissions from 1850 to present in the community earth system model. *Biogeosciences*, 13(11):3397–3426, jun 2016. doi: 10.5194/bg-13-3397-2016. URL https://doi.org/10.5194/bg-13-3397-2016.

Michele M. Rienecker, Max J. Suarez, Ronald Gelaro, Ricardo Todling, Julio Bacmeister, Emily Liu, Michael G. Bosilovich, Siegfried D. Schubert, Lawrence Takacs, Gi-Kong Kim, Stephen Bloom, Junye Chen, Douglas Collins, Austin Conaty, Arlindo da Silva, Wei Gu, Joanna Joiner, Randal D. Koster, Robert Lucchesi, Andrea Molod, Tommy Owens, Steven Pawson, Philip Pegion, Christopher R. Redder, Rolf Reichle, Franklin R. Robertson, Albert G. Ruddick, Meta Sienkiewicz, and Jack Woollen. MERRA: NASA's Modern-Era Retrospective Analysis for Research and Applications. *J. Clim.*, 24(14):3624–3648, July 2011. ISSN 0894-8755.

Donna B. Schwede and Gary G. Lear. A novel hybrid approach for estimating total deposition in the United States. *Atmos. Environ.*, 92:207–220, August 2014. ISSN 1352-2310.

Elena Shevliakova, Stephen W. Pacala, Sergey Malyshev, George C. Hurtt, P. C. D. Milly, John P. Caspersen, Lori T. Sentman, Justin P. Fisk, Christian Wirth, and Cyril Crevoisier. Carbon cycling under 300 years of land use change: Importance of the secondary vegetation sink. *Global Biogeochem. Cycles*, 23(2):GB2022, 2009. ISSN 1944-9224.

Samuel M. Simkin, Edith B. Allen, William D. Bowman, Christopher M. Clark, Jayne Belnap, Matthew L. Brooks, Brian S. Cade, Scott L. Collins, Linda H. Geiser, Frank S. Gilliam, Sarah E. Jovan, Linda H. Pardo, Bethany K. Schulz, Carly J. Stevens, Katharine N. Suding, Heather L. Throop, and Donald M. Waller. Conditional vulnerability of plant diversity to atmospheric nitrogen deposition across the United States. *Proc. Natl. Acad. Sci. U.S.A.*, 113(15):4086–4091, April 2016. ISSN 0027-8424, 1091-6490. doi: 10.1073/pnas.1515241113. URL `http://www.pnas.org/content/113/15/4086`.

D Simpson, H Fagerli, J Jonson, S Tsyro, P Wind, and JP Tuovinen. The emep unified eulerian model. model description. emep msc-w report 12003. *The Norwegian Meteorological Institute, Oslo, Norway*, 2003.

R. Singles, M.A. Sutton, and K.J. Weston. A multi-layer model to describe the atmospheric transport and deposition of ammonia in great britain. *Atmospheric Environment*, 32(3):393–399, feb 1998. doi: 10.1016/s1352-2310(97)83467-x. URL `https://doi.org/10.1016/s1352-2310(97)83467-x`.

B. Smith, D. Wårlind, A. Arneth, T. Hickler, P. Leadley, J. Siltberg, and S. Zaehle. Implications of incorporating n cycling and n limitations on primary production in an individual-based dynamic vegetation model. *Biogeosciences*, 11(7):2027–2054, apr 2014. doi: 10.5194/bg-11-2027-2014. URL `https://doi.org/10.5194/bg-11-2027-2014`.

J. Storkey, A. J. Macdonald, P. R. Poulton, T. Scott, I. H. Köhler, H. Schnyder, K. W. T. Goulding, and M. J. Crawley. Grassland biodiversity bounces back from long-term nitrogen addition. *Nature*, 528(7582):401–404, December 2015. ISSN 0028-0836. doi: 10.1038/nature16444. URL `http://www.nature.com/nature/journal/v528/n7582/abs/nature16444.html`.

M.A. Sutton, E. Nemitz, J.W. Erisman, C. Beier, K. Butterbach Bahl, P. Cellier, W. de Vries, F. Cotrufo, U. Skiba, C. Di Marco, S. Jones, P. Laville, J.F. Soussana, B. Loubet, M. Twigg, D. Famulari, J. Whitehead, M.W. Gallagher, A. Neftel, C.R. Flechard, B. Herrmann, P.L. Calanca, J.K. Schjoerring, U. Daemmgen, L. Horvath, Y.S. Tang, B.A. Emmett, A. Tietema, J. Peñuelas, M. Kesik, N. Brueggemann, K. Pilegaard, T. Vesala, C.L. Campbell, J.E. Olesen, U. Dragosits, M.R. Theobald, P. Levy, D.C. Mobbs, R. Milne, N. Viovy, N. Vuichard, J.U. Smith, P. Smith, P. Bergamaschi, D. Fowler, and S. Reis. Challenges in quantifying biosphere–atmosphere exchange of nitrogen species. *Environmental Pollution*, 150(1):125–139, nov 2007. doi: 10.1016/j.envpol.2007.04.014. URL `https://doi.org/10.1016/j.envpol.2007.04.014`.

Mark A. Sutton, David Simpson, Peter E. Levy, Rognvald I. Smith, Stefan Reis, Marcel Van Oijen, and Wim De Vries. Uncertainties in the relationship between atmospheric nitrogen deposition and forest carbon sequestration. *Glob. Chang. Biol.*, 14(9):2057–2063, September 2008. ISSN 1365-2486.

Pamela H. Templer, Kathleen C. Weathers, Amanda Lindsey, Katherine Lenoir, and Lindsay Scott. Atmospheric inputs and nitrogen saturation status in and adjacent to Class I wilderness areas of the northeastern US. *Oecologia*, 177(1):5–15, November 2014. ISSN 0029-8549, 1432-1939.

Elise M. Tulloss and Mary L. Cadenasso. Nitrogen deposition across scales: hotspots and gradients in a california savanna landscape. *Ecosphere*, 6(9):art167, sep 2015. doi: 10.1890/es14-00440.1. URL `https://doi.org/10.1890/es14-00440.1`.

Massimo Vieno, Anthony J. Dore, Peter Wind, Chiara Di Marco, Eiko Nemitz, Gavin Phillips, Leonor Tarrasón, and Mark A. Sutton. Application of the EMEP unified model to the UK with a horizontal resolution of 5x 5 km2. In *Atmospheric Ammonia*, pages 367–372. Springer Netherlands, 2009. doi: 10.1007/978-1-4020-9121-6_21. URL `https://doi.org/10.1007/978-1-4020-9121-6_21`.

K. C. Weathers, G. M. Lovett, G. E. Likens, and R. Lathrop. The Effect of Landscape Features on Deposition to Hunter Mountain, Catskill Mountains, New York. *Ecol. Appl.*, 10(2):528–540, 2000. ISSN 1051-0761. doi: 10.2307/2641112. URL `http://www.jstor.org/stable/2641112`.

Kathleen C. Weathers, Samuel M. Simkin, Gary M. Lovett, and Steven E. Lindberg. Empirical Modeling of Atmospheric Deposition in Mountainous Landscapes. *Ecol. Appl.*, 16(4):1590–1607, August 2006. ISSN 1939-5582.

G. R. Wentworth, J. G. Murphy, K. B. Benedict, E. J. Bangs, and J. L. Collett Jr. The role of dew as a night-time reservoir and morning source for atmospheric ammonia. *Atmos. Chem. Phys.*, 16(11):7435–7449, June 2016. ISSN 1680-7324. doi: 10.5194/acp-16-7435-2016. URL `http://www.atmos-chem-phys.net/16/7435/2016/`.

Jason J. Williams, Serena H. Chung, Anne M. Johansen, Brian K. Lamb, Joseph K. Vaughan, and Marc Beutel. Evaluation of atmospheric nitrogen deposition model performance in the context of u.s. critical load assessments. *Atmospheric Environment*, 150:244–255, feb 2017. doi: 10.1016/j.atmosenv.2016.11.051. URL `https://doi.org/10.1016/j.atmosenv.2016.11.051`.

Wei Wu and Charles T. Driscoll. Impact of climate change on three-dimensional dynamic critical load functions. *Environ. Sci. Technol.*, 44(2):720–726, 2010. PMID: 20020745.

Zhiyong Wu, Donna B. Schwede, Robert Vet, John T. Walker, Mike Shaw, Ralf Staebler, and Leiming Zhang. Evaluation and intercomparison of five north american dry deposition algorithms at a mixed forest site. *Journal of Advances in Modeling Earth Systems*, jun 2018. doi: 10.1029/2017ms001231. URL `https://doi.org/10.1029/2017ms001231`.

S. Zaehle, A. D. Friend, P. Friedlingstein, F. Dentener, P. Peylin, and M. Schulz. Carbon and nitrogen cycle dynamics in the o-CN land surface model: 2. role of the nitrogen cycle in the historical terrestrial carbon balance. *Global Biogeochemical Cycles*, 24(1):n/a–n/a, feb 2010. doi: 10.1029/2009gb003522. URL `https://doi.org/10.1029/2009gb003522`.

Leiming Zhang, Michael D. Moran, Paul A. Makar, Jeffrey R. Brook, and Gong Sunling. Modelling gaseous dry deposition in aurams: a unified regional air-quality modelling system. *Atmos. Environ.*, 36:537—560, 2002.

L. Zhu, D. Henze, J. Bash, G.-R. Jeong, K. Cady-Pereira, M. Shephard, M. Luo, F. Paulot, and S. Capps. Global evaluation of ammonia bidirectional exchange and livestock diurnal variation schemes. *Atmos. Chem. Phys.*, 15(22):12823–12843, November 2015. ISSN 1680-7324. doi: 10.5194/acp-15-12823-2015. URL `http://www.atmos-chem-phys.net/15/12823/2015/`.

---

## Author Response (AR2)

We wish to thank both reviewers for their comments

**1 Reply to the comments of reviewer 1**

1. **Line 122-123: I appreciate the revisions to the text made to try to address previous comments on this section. I still feel that the text is not particularly clear. How does the current state of the vegetation determine the vegetation type**
   We clarified that the state of the vegetation refers to its biomass. For instance, trees will only be able to develop over a certain biomass as described in Shevliakova et al. [2009].

2. **Line 135: Add "is" before "removed". I appreciate the additional information in the response to comments on this section. Is there a reference that supports your assumption?**

   We find that a high removal intensity is needed to prevent the growth of trees on pastures in tropical regions. The motivation for this choice has been clarified in the revised manuscript as follows:
   *For pasture, we assume that 10% of leaf biomass removed daily by grazing, provided LAI exceeds 2 to avoid overgragzing. This higher grazing frequency and intensity are needed to avoid the excessive growth of vegetation biomass on pasture in the tropics and mid latitudes, a problem which was noted in previous versions of LM3 [Malyshev et al., 2015] leading to misclassification of pasture vegetation cover as forests [Malyshev et al., 2015].*

   Note that we had mistakenly quoted the default removal intensity in LM3 (25%). Our simulations use a removal intensity of 10%.

3. **Line 155: I haven't seen the SAI included in Erisman's parameterization for Rac. Since the parameterization is based on a fit to data, does the inclusion of SAI violate that relationship?**
   First, $R_{ac,g}$ and $R_{ac,v}$ were expressed as conductances instead of resistances (as implied by equation 1). This has been corrected. On an annual basis, SAI is generally less than 15% of LAI, such that we expect little impact on the fit. We have clarified that SAI was not included in the expression derived by Erisman [1994].

   *Note that unlike Erisman [1994], we include SAI in the calculation of $R_{ac,g}$, which tends to reduce deposition to the ground in winter.*

4. **Line 157: Rac,v is not commonly used in models. It would help to explain this conceptually to clarify what it represents.**

   The use of Rac,v is required to account for the different turbulent regime that exists between canopy air and the vegetation and between the atmosphere and the canopy (Ra). However, we find that Rac,v is generally small relative to Ra. We have clarified that Rac,v and Rac,g designate the aerodynamic resistances within the canopy.

5. **Lines 155 − 199: Please specify the units for all parameters.**

   We have specified the units for the different parameters.

6. **Line 180: A description of the mesophyll resistance should appear near this section of the manuscript.**

   We have added the following text:
   *The mesophyll resistance is expressed following Wesely [1989]:*

   $$R_m(X) = \left(10^5/3000 \cdot \alpha(X) + 100 \cdot \beta(X)\right)^{-1} \tag{1}$$

7. **Line 192: Consider including the explanation in the response to comments rather than saying "to improve numerical stability" as it would be clearer.**

   We have modified the text as follows:
   *To avoid unrealistic oscillations in $v_d(\text{NH3})$ and $v_d(\text{SO}_2)$, we estimate..*

8. **Line 192: Is the 24-hour integrated deposition for the previous 24 hours?**

   Yes this is correct

9. **Line 209-210: The description of R2010_no_lu is not clear.**

   The text has been revised as follows:
   *An additional sensitivity experiment is performed (R2010_no_lu) in which natural vegetation is assumed to cover all vegetated tiles (i.e., no human land use)*

10. **Line 225: I appreciate the authors' attempt to address concerns about the text in this section by moving it. The text is better, but not quite there yet. Perhaps you are trying to say that there are still many aspects of deposition that poorly understood which results in uncertainties. By comparing your approach to others, you are not vastly different than the ensemble.**

    We agree with the reviewer that under dry condition and for temperatures close to 20°C, the different models simulate similar deposition velocities. This may not be surprising as they are all based on the Wesely scheme. However, the different models exhibit very different sensitivities to temperature and canopy wetness. The text has been revised follows:
    *Our comparison suggests that the implementation of the Wesely scheme in MOZART, AM3-LM3 DD, and GEOS-Chem produce similar $v_d(SO_2)$ and $v_d(NH_3)$ (within 50%) under dry conditions and for temperatures close to 20°C. However, differences in the sensitivity of $v_d(SO_2)$ and $v_d(NH_3)$ to environmental conditions (temperature, wetness, acidity) can result in large differences (>2). Such differences highlight the need for detailed evaluation of $v_d(X)$ across a wide range of conditions and chemical species [Wu et al., 2018].*

11. **Line 249: I don't understand how you sample from a diurnal cycle. Rather, I believe that you have paired the data in time and that you have analyzed the data for different parts of the diurnal cycle. Also, were you able to analyze morning dew period adequately with the time periods you selected. That time period can be particularly important for NH3 and SO2.**

    The reviewer is correct in its interpretation of our sampling strategy. The text has been clarified as follows:
    *We sample the simulated monthly $v_d(SO_2)$ at the location of the measurements in the tile that best represents the type of vegetation reported in the observations. When observations are available, we further distinguish between day-time and night-time as well as wet and dry conditions. For day-time and night-time observations, we sample the model from 8am to 5pm and 10pm to 4am, respectively. For wet conditions, we sample the model when the canopy wetness is greater than 10%.*

    LM3 does simulate dew, which is responsible for the increase in the wet cuticle conductance at night from 11pm to 6am at SOAS (Fig. 4). The text was revised as follows:
    *Finally, we note that the comparison against SOAS observations points to a significant high bias in simulated night-time deposition velocity. During this time period, the deposition is dominated by wet cuticles, which reflects the formation of dew in LM3.*

12. **Line 297: What do you think is driving the underestimation of Ra?**
    Our evaluation suggests that the model underestimates the stability of the nocturnal boundary layer. However, we have not yet found the root cause of this issue, which has implications beyond dry deposition (e.g., carbon cycle). The text was revised as follows:
    *Since this bias is found for all species including those with little surface resistance ($H_2O_2$ and $HNO_3$), it is likely to be associated with an underestimate of the stability of the nocturnal boundary layer.*

13. **Line 306: Consider rewording to "The grid-cell average dry deposition represents the area-weighted sum of the deposition fluxes"**

    We have followed the reviewer's suggestion

[revised manuscript text omitted]